# Spatially resolved analyses link genomic and immune diversity and reveal unfavorable neutrophil activation in melanoma

Akash Mitra [1,2,16], Miles C. Andrews [3,4,16], Whijae Roh [5], Marianna Petaccia De Macedo [6], Courtney W. Hudgens [6], Fernando Carapeto[6], Shailbala Singh[7], Alexandre Reuben [8], Feng Wang[1], Xizeng Mao[1], Xingzhi Song[1], Khalida Wani[6], Samantha Tippen[1], Kwok-Shing Ng [9], Aislyn Schalck[10], Donald A. Sakellariou-Thompson[11], Eveline Chen[6], Sangeetha M. Reddy[12], Christine N. Spencer[13], Diana Wiesnoski[1], Latasha D. Little[1], Curtis Gumbs[1], Zachary A. Cooper [14], Elizabeth M. Burton[3], Patrick Hwu[11], Michael A. Davies [11], Jianhua Zhang [1], Chantale Bernatchez[11], Nicholas Navin[10], Padmanee Sharma[15], James P. Allison[7], Jennifer A. Wargo [1,3], Cassian Yee[11,13], Michael T. Tetzlaff[6], Wen-Jen Hwu[11], Alexander J. Lazar [1,6] & P. Andrew Futreal [1✉]

Complex tumor microenvironmental (TME) features influence the outcome of cancer immunotherapy (IO). Here we perform immunogenomic analyses on 67 intratumor sub-regions of a PD-1 inhibitor-resistant melanoma tumor and 2 additional metastases arising over 8 years, to characterize TME interactions. We identify spatially distinct evolution of copy number alterations influencing local immune composition. Sub-regions with chromosome 7 gain display a relative lack of leukocyte infiltrate but evidence of neutrophil activation, recapitulated in The Cancer Genome Atlas (TCGA) samples, and associated with lack of response to IO across three clinical cohorts. Whether neutrophil activation represents cause or consequence of local tumor necrosis requires further study. Analyses of T-cell clonotypes reveal the presence of recurrent priming events manifesting in a dominant T-cell clonotype over many years. Our findings highlight the links between marked levels of genomic and immune heterogeneity within the physical space of a tumor, with implications for biomarker evaluation and immunotherapy response.

[1] Department of Genomic Medicine, University of Texas MD Anderson Cancer Center, Houston, Texas, USA. [2] Quantitative Sciences Graduate Training Program, Graduate School of Biomedical Sciences, Houston, Texas, USA. [3] Department of Surgical Oncology, University of Texas MD Anderson Cancer Center, Houston, Texas, USA. [4] Olivia Newton-John Cancer Research Institute and School of Cancer Medicine, La Trobe University, Heidelberg, VIC, Australia. [5] The Broad Institute of MIT and Harvard, Cambridge, Massachusetts, USA. [6] Department of Pathology, University of Texas MD Anderson Cancer Center, Houston, Texas, USA. [7] Department of Immunology, University of Texas MD Anderson Cancer Center, Houston, Texas, USA. [8] Department of Thoracic Head and Neck Medical Oncology, University of Texas MD Anderson Cancer Center, Houston, Texas, USA. [9] Institute for Personalized Cancer Therapy, University of Texas MD Anderson Cancer Center, Houston, Texas, USA. [10] Department of Genetics, University of Texas MD Anderson Cancer Center, Houston, Texas, USA. [11] Department of Melanoma Medical Oncology, University of Texas MD Anderson Cancer Center, Houston, Texas, USA. [12] Department of Breast Medical Oncology, University of Texas MD Anderson Cancer Center, Houston, Texas, USA. [13] Parker Institute for Cancer Immunotherapy, San Francisco, California, USA. [14] AstraZeneca, Gaithersburg, Maryland, USA. [15] Department of Genitourinary Medical Oncology, University of Texas MD Anderson Cancer Center, Houston, Texas, USA. [16]These authors contributed equally: Akash Mitra, Miles C. Andrews.
✉email: AFutreal@mdanderson.org

Modern treatment paradigms increasingly expose patients with metastatic melanoma to multiple treatment modalities through the course of their disease[1]. Immune checkpoint blockade in particular has revolutionized the therapeutic landscape, yet durable clinical benefit remains limited to a subset of patients[2,3]. Numerous biomarker studies aiming to elucidate why the majority of patients fail to respond have revealed both immune and genomic contributors to therapeutic activity[3–6], but incorporation of such factors into clinical practice is not yet routine.

Intra- and inter-tumoral heterogeneity can influence lesion-specific and overall patient response to therapy[2,7], and may contribute significantly to tumor-immune evasion[2,8]. Studying the influence of intratumoral heterogeneity (ITH) using standard approaches such as bulk tumor sequencing or single-cell sequencing generally loses spatial information. Thus, here we perform spatially detailed immune and genomic analysis of three metastatic lesions, including 67 sub-regions of one tumor sampled throughout its entire mass, from a heavily treated but long-term surviving melanoma patient. Through molecular analyses coupled with strict retention of spatial detail, we reconstruct the striking relationship between genomic and immune heterogeneity. We identify a remarkable link between copy number gain of chromosome 7 and an unfavorable immune composition driven by neutrophil activation recapitulated within TCGA melanoma samples and dominating non-responders to checkpoint blockade immunotherapy across multiple published cohorts. We also identify a long-term persistent T-cell clonotype having potential relevance to vaccine exploration and cellular immunotherapy.

## Results

**Longitudinal tumor sampling.** Tumor and blood biospecimens were obtained from a Caucasian female diagnosed with de novo stage IV M1b melanoma of unknown primary metastatic to the left lung at the age of 77 years. Following initial curative intent wedge resection of the solitary $NRAS^{Q61R}$ mutated lung metastasis (Fig. 1a, lesion 1), her clinical course was remarkable for long-term survival despite multiple lines of therapy for widely distributed soft tissue metastases with limited to no objective response over the following 8 years (Fig. 1a). To explore the relevance of ITH to the setting of long-term survival with metastatic disease, we studied a ventral abdominal wall metastasis resected due to isolated progression during therapy with the PD-1 inhibitor pembrolizumab. This mass (Fig. 1a, lesion 2) was subjected to extensive multidimensional spatial and immunogenomic profiling by serial sectioning and the use of alternate tumor sections for region-matched immunohistochemistry (IHC) analyses (odd-numbered slices) and genomic and proteomic analyses (even-numbered slices; Fig. 1b). Individual sections were further sub-divided into 20 regions (Fig. 1b and Supplementary Fig. 1), producing a total of 67 regions assessed by at least one analytical platform (Supplementary Data 1).

**Mutational ITH is highly prevalent and spatially restricted.** To characterize genomic ITH within the tumor specimen progressing during PD-1 inhibitor treatment ("on-PD-1 inhibitor" tumor), we performed deep targeted DNA sequencing for a panel of 265 cancer-related genes (Supplementary Data 2) of DNA from 41 tumor sub-regions. Of 53 identified somatic mutations, 28% (15 of 53) were shared in all 41 regions whereas 30% (16 of 53) were restricted to a single region (Supplementary Data 3), consistent with a degree of mutational ITH not previously described at this resolution. Somatic mutations in putative melanoma driver genes including $NRAS^{Q61R}$, $BRAF^{G421R}$ and $MAP2K1^{P124S}$, all key components of the mitogen-activated protein kinase

(MAPK) pathway, were ubiquitously detected in all 41 regions, supporting the notion that somatic mutational heterogeneity is predominantly attributable to passenger mutations. A $JAK1^{P1044S}$ mutation affecting the activation loop of JAK1 that was detected in all 41 regions and conferred signaling hypomorphism by Ba/F3 mutant transformation assay (Fig. 1c) potentially contributed to the immunotherapy resistance displayed by this tumor clinically[9].

**Genomic ITH is dominated by copy number alterations.** Analysis of copy number alterations (CNAs) detected across all 41 deeply sequenced samples identified gains of chromosome 6p and 20q, and losses of chromosome 6q and 9p, each of which has previously been identified in melanoma clinical samples (Fig. 1d)[8]. Subclonal alterations were also seen, including chromosome 7 gain in four samples, whole-chromosome 10 loss in five samples, 10p loss in one sample, and chromosome 13 gain in four samples. Samples with subclonal loss of chromosome 10 were localized in adjacent tumor slices, but subclonal gains of chromosome 7 and 13 were found at non-contiguous sites (Fig. 1d). Although previous studies have shown metastatic potential being associated with the loss of chromosome 10[10], we found evidence of regional losses of chromosome 10, most extensively along the tumor margin, suggesting this may be selected for in the context of stromal interactions at advancing tumor margins. Nearly half (17/39, 44%) of the differentially expressed genes associated with chromosome 10 copy number losses were located on chromosome 10 itself, characterized by relatively high expression but low fold change. Additional, more pronounced changes (at fold-change level) were observed in differentially expressed genes located on other chromosomes, such as $MT1B$ (chr16), $TNNT3$ (chr11), $MUC12$ (chr7), and $RPS6KA6$ (chrX) (Supplementary Data 4). In addition, unique chromosomal CNAs were found in nearly all (12 of 14) regions, demonstrating that CNAs may develop along spatially distinct trajectories even within a single metastasis. Comparing CNAs across longitudinal metastases of this patient, we also identified progressive stepwise regional loss of chromosome 10 in relation to therapy (pre-, on-, and post-PD-1 inhibitor therapy), thus implicating this CNA in both tumor margin dynamics and overall disease progression (Supplementary Fig. 2).

**Immune cell content is highly and spatially diverse.** We next characterized the ITH of gene expression patterns in the tumor, to gain insight into the nature of local tumor-immune micro-environments. Unexpectedly, unsupervised hierarchical clustering based on transcriptomic profiling revealed limited association between regional gene expression and histologic features such as intratumoral site (e.g., "core" surrounded only by tumor mass vs. "margin" spanning the tumor edge and including surrounding tissue) (Fig. 2a and Supplementary Fig. 3A). We then used several immune deconvolution tools to enumerate separate immune, stromal, and tumor cell populations, as well as melanoma-, AXL-, and MITF-related gene expression programs[11–16]. Sub-regions with high content of one immune cell subset generally displayed an enrichment for multiple cell subsets, indicative of a broadly diverse infiltrating immune population, and consistent with the observed high correlations between immune cell marker stains by IHC (Fig. 2b, c and Supplementary Fig. 3B-C). Samples with higher immune activity were over-represented at tumor margin sites ($p < 0.001$, Fisher's exact test), reflecting the spatially excluded (i.e., peri-tumoral) leukocytic accumulation observed on IHC (Fig. 2a and Supplementary Fig. 1). A notable exception was particularly high T- and B-cell signatures in multiple samples of section 8 (8A6, 8A7, 8A8, and 8A13), which was highly necrotic and displayed heavy neutrophil infiltration on matched formalin-fixed paraffin-embedded (FFPE) slices (CD15 stain; Supplementary Fig. 3D),

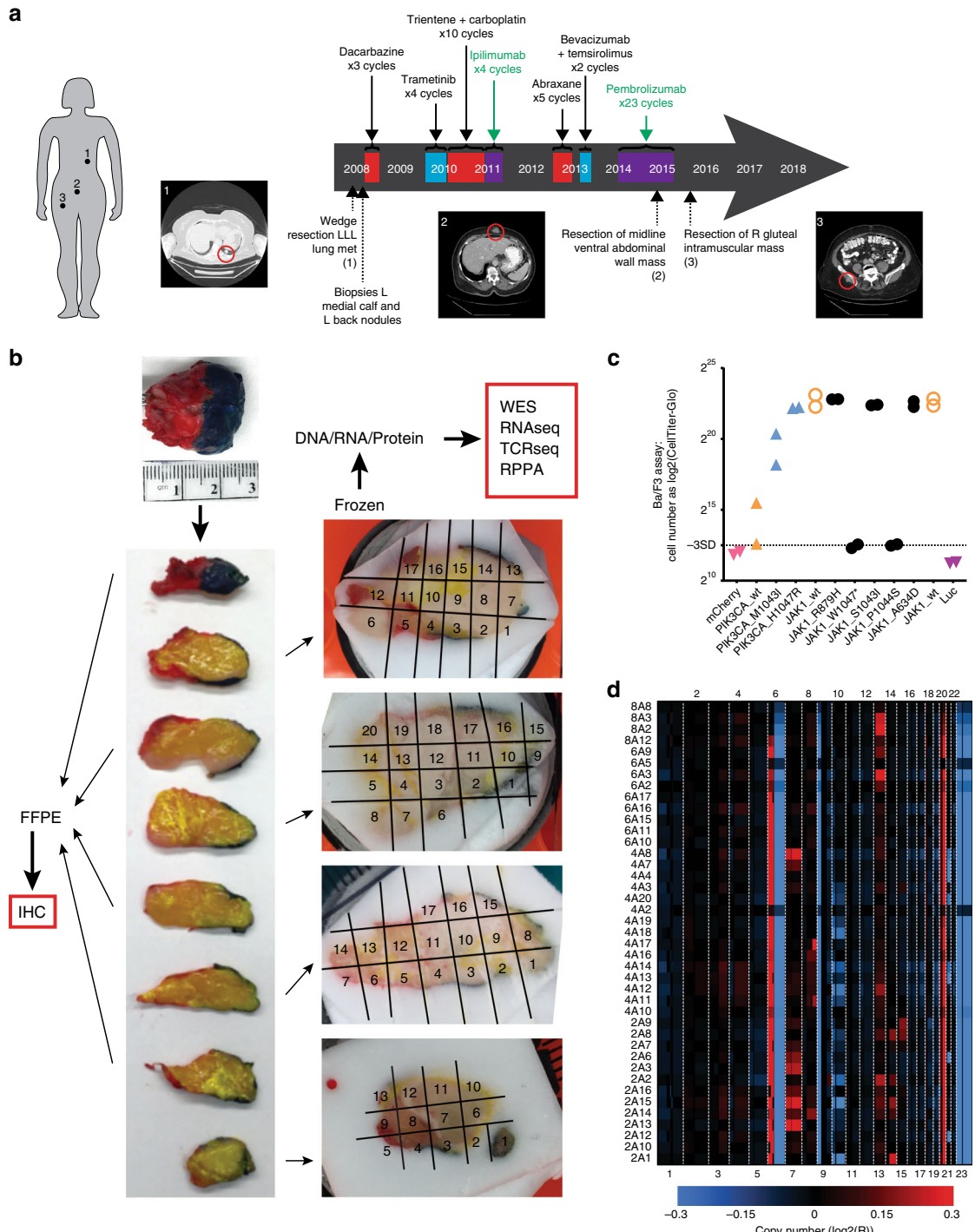

**Fig. 1 Genomic inter- and intratumoral heterogeneity in a heavily treated melanoma patient are driven by copy number alterations. a** Timeline of treatments and surgical sampling of three distinct melanoma tumors from a long-term surviving patient with largely treatment unresponsive metastatic melanoma. Treatment modality is indicated by color (red, chemotherapy; blue, targeted therapy; purple, immunotherapy). Molecularly profiled lesions are indicated: index left lower lobe (LLL) lung metastasis (lesion 1), progressing ventral abdominal wall mass (lesion 2), and slowly progressing right gluteal mass (lesion 3). **b** Sectioning and use of the on-PD-1 inhibitor abdominal wall lesion. The tumor was oriented by lateral inking (red, left; blue, right), sliced, and laid on a grid. The odd-numbered slices were processed for FFPE and used for immunohistochemistry, whereas the even-numbered slices were fresh-frozen and used for genomic and proteomic analyses (whole exome sequencing (WES), RNA sequencing, TCR sequencing, reverse-phase protein array (RPPA)). **c** Functional hypomorphism of the identified *JAK1* mutation (*JAK1*[P1044S]) was identified by Ba/F3 transformation assay. Also shown are known oncogenic *JAK1* variants (*JAK1*[R879H], *JAK1*[S1043I], *JAK1*[A634D]), wild-type *JAK1*, a truncating *JAK1* hypomorph (*JAK1*[W1047*]), and oncogenic *PIK3CA* variants. **d** Copy number alterations in each region of the tumor are shown in the chromosome coordinate as log2-transformed copy number probe intensities R (observed intensity/reference intensity); copy number gains are shown as red and copy number losses as blue.

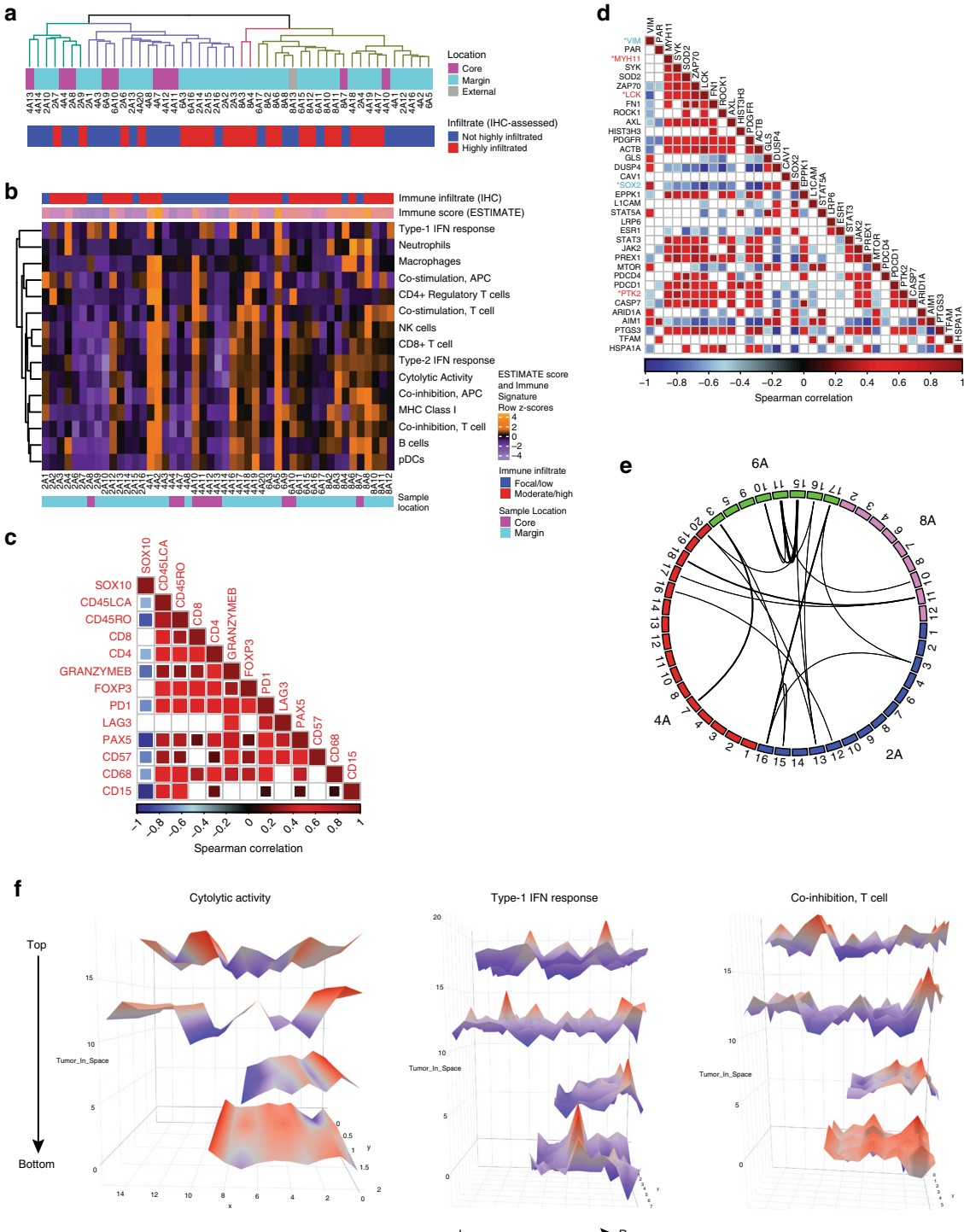

although this could not be determined as the cause or consequence of necrosis. In many cases, high immune cell signatures were accompanied by high expression of interferon-stimulated genes. Relatively low levels of variability were observed in expression of genes linked to melanoma cell phenotype (i.e., melanocytic, MITF-related, and AXL-related gene sets); however, the same samples that displayed prominent and uniform immune signature upregulation also displayed a non-melanocytic phenotype that is known to be associated with mesenchymal-like and pro-invasive cellular behaviors, evidenced by high expression of the AXL-related gene set, and correspondingly low expression of the melanocytic and MITF-related gene sets (Supplementary Fig. 3A)[17,18].

Mirroring transcriptional ITH, a high degree of ITH was observed between samples at the protein level, measured by a 296-target reverse-phase protein array (Supplementary Fig. 3E). Within the most spatially variable proteins, two notable modules of co-expressed proteins emerged; one including AIM1, ARID1A, MTOR, STAT5A, DUSP4, and SOX2, resembling a melanocytic cellular origin and an anti-correlated set comprising AXL, PDGFR, JAK2, STAT, PDCD1 (PD-1), and PREX1, suggesting a mesenchymal-shifted and/or immune-infiltrated set (Fig. 2d). Several proteins were significantly associated with either locally low (VIM, SOX2) or high (MYH11, LCK, PTK2) immune infiltrate (false discovery rate (FDR) < 0.10, Spearman's $\rho$ rank correlation, Fig. 2d).

**Fig. 2 Immune-driven transcriptional heterogeneity implicates diverse immune cell populations and highly localized immune activation or suppression.**
**a** Unsupervised hierarchical clustering based on the top 1000 most variant (mean absolute deviation) genes across all samples of the on-PD-1 inhibitor tumor, demonstrating limited associations between clustering of samples and tumor location based on the general transcriptome, but apparent association between high immune infiltrate and location at the tumor margin. **b** Heatmap of immune signature gene sets (from Rooney et al.[13]) across tumor sub-regions demonstrate dispersed pockets of immune activation or suppression throughout the tumor, wherein immune-high samples (e.g., 4A1/4A2, 6A5, and 8A6) are physically distant from each other within the tumor mass. IHC-based immune infiltrate and ESTIMATE immune scores (top) and IHC-based tumor sample location (bottom) are indicated. **c** Immunohistochemical marker inter-correlations demonstrating generally diverse representation of immune cell types when infiltrates are present. Data are Spearman's correlation values (with Benjamini–Hochberg correction; only $p < 0.05$ are shown) indicated according to the color scale shown. **d** Correlation of most variably abundant proteins measured by reverse-phase protein array, revealing two main modules of highly correlated molecules. Proteins displaying statistically significant (FDR $p < 0.10$) correlation with immune infiltrate are indicated by * and color (blue = anti-correlated with immune infiltrate, red = directly correlated with immune infiltrate). Data are Spearman's correlation values (with Benjamini–Hochberg correction; only $p < 0.05$ are shown) indicated according to the color scale shown. **e** Sample-wide similarity of immune activity was estimated by calculation of the distance matrix between samples using the immune activation signature expression data; lines connect samples in the top quartile of similarity scores, demonstrating global immune signature similarities that are not restricted by intratumoral location. **f** Three-dimensional spatial mapping of subregion Cytolytic activity signature, Type-I IFN response signature and Co-inhibition, T-cell signature scores derived from transcriptomic data, in the manner of Rooney et al.[13]. Data map the geometric mean of genes included in each gene set onto three-dimensional space representing the tumor slices, with color and height indicating expression value (higher expression = red peaks, lower expression = blue troughs).

**Sites of similar immune composition may be spatially remote.** Using sample-wide Euclidean distance metrics to connect samples with highly similar immune composition based on immune deconvolution rather than reductive immune scores or overall immune cell densities, we found that similar immunophenotypes were unrestricted by location at core or margin sites (e.g., core 4A7 vs. margin 6A3), or by spatial proximity (Fig. 2e). Three-dimensional mapping across all sampled regions of the tumor revealed clear but disconnected pockets of immune activation and suppression as typified by signatures derived for cytolytic activity, type-I intereron (IFN) activity and an anti-inflammatory signature (Fig. 2f)[13], indicative of a degree of immunophenotype convergence. To address the implications of regional immune phenotype variation for clinical biomarker assessment, we performed consensus clustering of samples based on gene expression data and identified an optimal four cluster solution, being the minimum number of distinct regional "phenotypes" that would need to be sampled, to approximately represent the transcriptional heterogeneity present across the entire tumor mass (Supplementary Fig. 3F). Importantly, we found that each of these clusters contained non-contiguous samples, indicating a low chance of serendipitously sampling all microenvironmental types with any single biopsy of the lesion (Supplementary Fig. 3F).

**ITH implicates simultaneous methods of immune exclusion.** Given progression of this tumor through anti-PD-1 immunotherapy and previous findings suggesting a predictive significance of the immune status at the invasive tumor margin[19], we next compared tumor regions having either a high or low immune cell content as measured by a pan-leukocyte stain (CD45-LCA) on IHC (Supplementary Fig. 1). The most differentially expressed genes enriched in heavily infiltrated subregions included *FCRL1*, *CADM3*, *CR2*, and *PAX5* (Fig. 3a, Table 1, and Supplementary Data 5), as well as genes involved in T-cell function including *CD3D*, *CD28*, *ZAP70*, and *CD40LG*, in agreement with extensive CD8 and CD4 staining of mononuclear cells within these highly immune-infiltrated regions by IHC (Fig. 3b–d and Supplementary Data 5). At the Gene Ontology (GO) pathway level, these differentially expressed genes contributed to mixed T- and B-cell enrichments, and a substantial degree of functional gene connectivity (Fig. 3c, d), which was maintained even when specifically comparing samples located at the tumor margin (Supplementary Fig. 4A-B). We also identified a clear B-cell gene enrichment in highly immune-infiltrated samples, driven by *PAX5*, *BLK*, *CD19*, *CLECL1*, *CD180*, *CD22*, and *CD79A* (Fig. 3c, d, Table 2, and Supplementary Data 5).

Parallel PAX5 immunostaining of tumor sections confirmed B-cell lineage presence within these immune-infiltrated samples localized to intra- and peri-tumoral leukocytic infiltrates, or within dense para-tumoral clusters associated with blood vessels and other immune cell types, suggestive of tertiary lymphoid structures (Fig. 3e). In addition, a pro-tumorigenic M2 macrophage signature was evident throughout most regions of the tumor (Supplementary Data 6)[16,20]. Tumor-associated macrophages at the tumor periphery are known to be associated with tumor progression[21]; thus, these data implicate active participation of immunosuppressive macrophages in the observed clinical progression of this tumor despite anti-PD-1 therapy.

Reasoning that grouped analyses may obscure the true extent of variability in gene expression between individual samples, we performed single-sample gene set enrichment analysis (ssGSEA) to gain a finer resolution of the functional transcriptomic activity[22]. Strikingly, unsupervised hierarchical clustering of the samples based on ssGSEA of Hallmark gene sets again revealed little similarity in terms of physical location within the tumor or the extent of peri-/intratumoral immune infiltrate (Fig. 3f). Samples with prominent enrichment of WNT/β-catenin signaling (2A10, 2A13, 2A16, 4A11, and 8A4), which is a known tumor cell-intrinsic mechanism of immune cell exclusion[23], were typically located at the tumor margin but did not show consistent association with immune cell content, although when immune cells were present, they were largely peri- or extra-tumoral in distribution. These data suggest that WNT/β-catenin signaling may contribute to exclusion of an immune infiltrate when one is present, but additional factors are necessary to explain the complete absence of an immune infiltrate from some regions. Intriguingly, despite known presence of activating *NRAS* and *MAP2K1* mutations, phospho-ERK1/2 (pERK) expression (by IHC) was largely absent from tumor cells, except when located at or immediately beneath the tumor margin (Fig. 3g), suggesting MAPK activation in response to factors originating near the tumor surface. Areas of strong tumor cell pERK staining were frequently observed in association with overlaid peri-tumoral immune infiltrates; thus, tumor cell ERK activation may be actively involved in the maintenance of immune cell exclusion, and at a scale that is significantly more spatially localized than previously thought based on pre-clinical models and broad assessments of patient samples[24,25].

**Integrative analyses of multimodal molecular phenotypic data.** Having identified clear links between immune and genomic heterogeneity throughout sub-regions of this tumor, we sought to

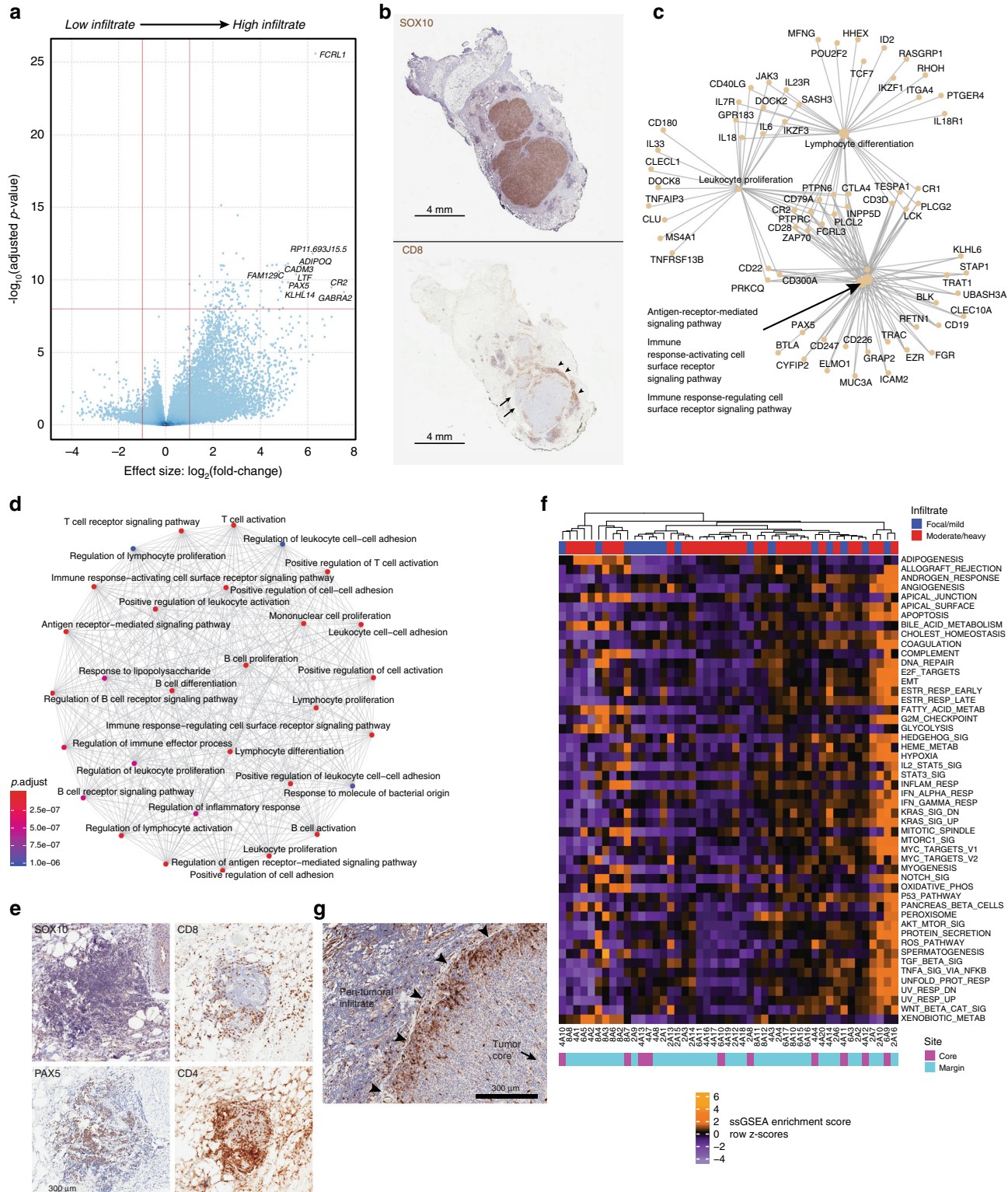

identify genomic features underlying this through an integrative analysis of CNA and mRNA data, with the addition of methylation as a potential modulator of transcriptional activity. Examining the 560 most variably expressed genes for which all genomic data were available, unlike CNA, underlying methylation patterns appeared more variable between samples rather than between genes, implying that a genome-wide methylation state interacts with more localized genomic and posttranscriptional influences to affect gene expression in this context (Fig. 4a).

From an immune standpoint, transcriptome-derived ESTIMATE immune scores trended lower in regions with chromosome 10 losses ($p = 0.088$, two-sided $t$-test) and were significantly lower in regions with subclonal gain of chromosome 7 ($p = 0.018$, two-sided $t$-test)[12]. Similarly, differentially expressed genes were enriched within CNA-affected regions of chromosome 6 and 7 ($p = 3.53e − 7$ and $1.22e − 5$, respectively, Benjamini–Hochberg corrected $p$-value). As expected, the clear majority of genes on chromosome 7 showed positive correlations between copy

**Fig. 3 Transcriptomic analysis demonstrates considerable ITH underlying convergent immune phenotypes. a** Volcano plot of differentially expressed genes comparing high vs. low immune infiltrate regions across the tumor core and margins. Vertical red lines indicate a minimum twofold change in expression value; horizontal red line indicates the adjusted *p*-value threshold of <1e-8. **b** Representative IHC sections demonstrating matched tumor content (SOX10 stain, above) and immune infiltration (CD8 stain, below) illustrating substantial variation of local CD8 T-cell content ranging from low (arrows) to high (arrowheads). **c** Gene connection network of genes upregulated in immune-infiltrated samples that contribute to highly enriched GO terms/pathways, showing substantial connectivity. **d** Functional enrichment network showing diverse representation of immune cell pathways and functions in the immune infiltrate-derived differentially expressed genes, dominated by highly inter-connected T and B-lymphocyte-related terms. **e** Representative IHC images of a para-tumoral lymphoid structure present in section 1B, demonstrating absence of tumor cells (SOX10) but mixed populations of CD4+, CD8+, and PAX5+ lymphocytes. Magnification ×10. **f** Single-sample gene-set enrichment analysis demonstrating spatially discontiguous enrichment of functionally relevant gene sets throughout the tumor. IHC-based estimate of immune infiltrate (top) and sample location (bottom) are indicated. **g** Immunostained tumor tissue revealing restriction of tumor cell phospho-ERK1/2 expression (brown staining) to cells located at or immediately subjacent to the tumor cell surface. Arrowheads: tumor-surrounding tissue interface. Magnification ×10.

number and transcript abundance, consistent with CNA representing a dominant method of regulation of these genes (Fig. 4b, upper panel). However, copy number vs. transcript correlations were negligible for four genes (*CALD1, CCT6A, CHCHD2*, and *ESYT2*) and negative for six genes (*ACTB, AEBP1, COL1A2, GIMAP4, GIMAP7*, and *SFRP4*), suggesting that additional mechanisms regulate transcript abundance of these genes, such as methylation. Gene methylation was inversely correlated with transcript abundance and thus consistent with negative regulation of transcription for most chromosome 7 genes (Fig. 4b, lower panel), including all but three of the copy number discordant genes (*GIMAP4, GIMAP7*, and *SFRP4*). Notably, *SFRP4* is a soluble modulator of Wnt signaling that may antagonize Wnt-driven immune exclusion when highly expressed. *GIMAP4* is known to be involved in the regulation of Th1 vs. Th2 T-cell phenotypes. Using the unique genome–phenotype associations of chromosome 7 to model overall regulatory complexity and ITH, we found strikingly little similarity in the unsupervised clustering patterns of sub-regions based on copy number, methylation, or transcript abundances, evidenced by a high degree of cross-cluster entanglement, indicating the action of additional (unmeasured) factors in regulation of these genes (Fig. 4c). Furthermore, sub-regions of similar immune cell content (measured by IHC) were generally dispersed throughout the clusters, thus demonstrating unequivocally the presence of profound ITH underlying broadly similar appearing immune microenvironments at the cellular level.

**Tumor-specific and agnostic T-cell recruitment occurs on a regional basis.** T-cell function is central to current immunotherapy efficacy; hence, we performed sequencing of the variable region of the T-cell receptor (TCR) β-chain using both DNA and RNA approaches to study T-cell ITH. T-cell repertoire clonality was substantially variable between samples (Fig. 4d), suggesting highly localized patterns of clonal expansion and contraction that result in variable repertoire composition throughout the volume of an individual tumor. Only 0.02% of all TCR rearrangements were detectable in all regions of the tumor and the vast majority (74.6%) were restricted to a single region. We identified the top five highly transcriptionally active T-cell clones per sample by plotting complementary productive frequencies generated from DNA- and RNA-based approaches (Supplementary Fig. 5): three dominant clonotypes were present at high abundances across multiple regions of the metastasis (Fig. 4e). One dominant clonotype, at the amino acid level, present as a top five clone in all samples (CSVPTSGSRDNEQFF), was most prevalent in the upper sections (2 and 4) and least prevalent in the lowest section 8, which also had the lowest proportion of viable tumor. The next two most prevalent clones present in 72% (CASSSLQGARREETQYF) and 69% (CASSLHGDQPQHF) of all samples were particularly enriched in sections 6 and 8.

We examined repertoire overlap between samples to infer intratumor trajectories of T-cell clones and found a high level of T-cell clonal overlap between samples within section 8 (Fig. 4f). Conversely, there was greater sharing of clonotypes between sections 4 and 6, and more sparsely with section 2, paralleled by evidence of greater immune activation in key regions of these sections. The greater restriction of T-cell clones within section 8 may reflect a functionally distinct T-cell repertoire reacting to the prevailing necrotic conditions seen histologically throughout much of this section. Overall, this spatial variation suggests underlying differences in regional immunogenicity and antigenicity driving local accumulation of different T-cell clones.

The observation of a marked T-cell repertoire ITH (Fig. 4d) and apparently distinct T-cell repertoires between distinct regions of the tumor was surprising given the comparatively similar mutational landscape between tumor sub-regions. To explore the relationship between T-cell clonal composition and tumor mutations, we compared the truncal set of 15 mutations found in every subsample of the metastasis with the most highly expanded TCR-Vβ sequences found simultaneously in high proportions across all regions. In general, the productive frequencies of these expanded T-cell clones correlated positively with the mutation variant allele frequencies (VAFs), suggesting a surrogate relationship between VAF, tumor content, and T-cell clones reactive to tumor (but not necessarily these exact mutations). However, some expanded T-cell clones showed inverse or mixed patterns of correlation with this set of truncal mutations, (Supplementary Fig. 6), including several clones negatively correlated with all the shared mutations, such as one (CASSLHGDQPQHF) that was predominantly found expanded in the necrotic slices. Together, these data indicate that although certain expanded T-cell clones correlate positively with a set of truncal tumor mutations and are likely tumor specific, a distinct population of expanded T-cell clones generally anti-correlate with truncal tumor mutations and, although present within some regions of the tumor, are likely not specific for tumor antigens and may be nonspecifically recruited into regions of inflamed/ necrotic TME.

**T-cell clone persistence reveals recurrent priming and functional diversity.** Leveraging the availability of peripheral blood samples and tumor samples obtained from distinct metastatic deposits spanning treatment-naive progression on-PD-1 inhibitor and progression post-PD-1 inhibitor time points (Fig. 1a), the temporal dynamics of the T-cell repertoire were evaluated. Strikingly, the dominant clone present within the progressing abdominal wall tumor during PD-1 inhibitor therapy was not only present over time but was the most hyperexpanded clone within the treatment-naive lung tumor sampled 7 years earlier (Fig. 4e). Evaluation of predicted neoantigens revealed one (*ZDHHC17* p.H507Y; IC50 = 77.17 nM) that was shared among

**Table 1 Differentially expressed genes confirm the activity of multiple immune subsets in regions of heavy immune infiltration.**

| Gene | Base mean | log$_2$ fold change | lfc SE | stat | *p*-Value | *p*-Adj |
|---|---|---|---|---|---|---|
| *RP11.280H21.1* | 2.22 | −4.34 | 1.65 | −2.63 | 8.54E − 03 | 4.95E − 02 |
| *RP11.376M2.2* | 3.45 | −3.93 | 1.12 | −3.50 | 4.63E − 04 | 5.46E − 03 |
| *AL109763.2* | 1.75 | −3.61 | 1.13 | −3.18 | 1.47E − 03 | 1.36E − 02 |
| *AC093850.1* | 3.34 | −3.51 | 1.18 | −2.97 | 2.96E − 03 | 2.29E − 02 |
| *RP11.114H23.3* | 2.91 | −3.43 | 0.97 | −3.52 | 4.34E − 04 | 5.16E − 03 |
| *CTD.2651C21.3* | 1.51 | −3.41 | 1.03 | −3.31 | 9.34E − 04 | 9.59E − 03 |
| *RP11.307L14.2* | 1.88 | −3.34 | 1.08 | −3.10 | 1.91E − 03 | 1.65E − 02 |
| *ICAM5* | 1.66 | −3.19 | 1.10 | −2.89 | 3.86E − 03 | 2.80E − 02 |
| *RP11.29P20.1* | 12.71 | −3.06 | 0.68 | −4.51 | 6.46E − 06 | 1.58E − 04 |
| *RNF208* | 1.39 | −2.89 | 1.07 | −2.70 | 6.99E − 03 | 4.30E − 02 |
| *IL1RAPL2* | 12.62 | −2.80 | 0.65 | −4.32 | 1.54E − 05 | 3.28E − 04 |
| *EFNA3* | 1.98 | −2.74 | 0.91 | −3.00 | 2.71E − 03 | 2.14E − 02 |
| *GPR115* | 6.36 | −2.59 | 0.69 | −3.76 | 1.67E − 04 | 2.42E − 03 |
| *MAST1* | 9.13 | −2.58 | 0.60 | −4.33 | 1.50E − 05 | 3.22E − 04 |
| *RNU6.850 P* | 2.17 | −2.57 | 0.91 | −2.82 | 4.79E − 03 | 3.27E − 02 |
| *RP11.191L17.1* | 5.79 | −2.53 | 0.91 | −2.79 | 5.31E − 03 | 3.53E − 02 |
| *AC007091.1* | 3.29 | −2.40 | 0.88 | −2.71 | 6.68E − 03 | 4.17E − 02 |
| *RP11.67M1.1* | 6.98 | −2.39 | 0.58 | −4.15 | 3.39E − 05 | 6.41E − 04 |
| *LINC00919* | 3.72 | −2.33 | 0.84 | −2.77 | 5.60E − 03 | 3.67E − 02 |
| *AC018742.1* | 16.53 | −2.33 | 0.80 | −2.91 | 3.59E − 03 | 2.66E − 02 |
| *SMYD1* | 4.20 | −2.29 | 0.68 | −3.35 | 8.14E − 04 | 8.65E − 03 |
| *RN7SL151P* | 4.88 | −2.23 | 0.77 | −2.88 | 3.99E − 03 | 2.86E − 02 |
| *RLBP1* | 27.57 | −2.09 | 0.43 | −4.82 | 1.46E − 06 | 4.65E − 05 |
| *CTD.3064H18.4* | 8.82 | −2.01 | 0.70 | −2.86 | 4.20E − 03 | 2.97E − 02 |
| *NDUFAF4P3* | 9.37 | −2.00 | 0.46 | −4.40 | 1.11E − 05 | 2.49E − 04 |
| *RP1.153P14.5* | 3.39 | 5.48 | 1.20 | 4.58 | 4.74E − 06 | 1.22E − 04 |
| *LTF* | 351.40 | 5.52 | 0.73 | 7.60 | 2.89E − 14 | 4.03E − 11 |
| *LINC00086* | 6.09 | 5.52 | 1.26 | 4.39 | 1.11E − 05 | 2.50E − 04 |
| *IGLV3.21* | 18.88 | 5.62 | 0.91 | 6.20 | 5.73E − 10 | 8.57E − 08 |
| *ADH1B* | 128.01 | 5.67 | 1.05 | 5.38 | 7.39E − 08 | 4.12E − 06 |
| *EPPK1* | 9.80 | 5.68 | 0.89 | 6.40 | 1.55E − 10 | 3.18E − 08 |
| *hsa.mir.5195* | 4.71 | 5.70 | 1.05 | 5.45 | 5.12E − 08 | 3.03E − 06 |
| *SLC16A9* | 6.97 | 5.73 | 1.16 | 4.92 | 8.62E − 07 | 3.03E − 05 |
| *ADIPOQ* | 300.86 | 5.73 | 0.73 | 7.86 | 3.72E − 15 | 9.45E − 12 |
| *RP11.89M16.1* | 4.91 | 5.75 | 1.00 | 5.76 | 8.42E − 09 | 6.92E − 07 |
| *KLHL14* | 19.27 | 5.83 | 0.82 | 7.10 | 1.23E − 12 | 5.96E − 10 |
| *PGBD4P1* | 5.38 | 5.90 | 1.05 | 5.62 | 1.96E − 08 | 1.37E − 06 |
| *MDS2* | 4.65 | 5.93 | 1.10 | 5.39 | 7.06E − 08 | 3.99E − 06 |
| *SAA2* | 11.85 | 5.99 | 0.97 | 6.17 | 6.69E − 10 | 9.56E − 08 |
| *DSC3* | 5.00 | 6.06 | 1.25 | 4.85 | 1.25E − 06 | 4.11E − 05 |
| *PCK1* | 26.46 | 6.19 | 1.08 | 5.75 | 9.17E − 09 | 7.40E − 07 |
| *RP11.693J15.5* | 42.65 | 6.21 | 0.76 | 8.16 | 3.42E − 16 | 1.52E − 12 |
| *TNNT3* | 9.87 | 6.22 | 1.22 | 5.10 | 3.31E − 07 | 1.40E − 05 |
| *FCRL1* | 50.74 | 6.33 | 0.55 | 11.53 | 9.37E − 31 | 2.50E − 26 |
| *MAL2* | 7.46 | 6.64 | 1.05 | 6.30 | 2.91E − 10 | 5.03E − 08 |
| *CAPN6* | 7.91 | 6.72 | 1.10 | 6.11 | 9.98E − 10 | 1.33E − 07 |
| *RBP4* | 9.54 | 6.72 | 1.18 | 5.72 | 1.07E − 08 | 8.49E − 07 |
| *CR2* | 574.56 | 7.01 | 0.97 | 7.23 | 4.99E − 13 | 3.24E − 10 |
| *MFSD2A* | 17.06 | 7.02 | 1.10 | 6.38 | 1.82E − 10 | 3.49E − 08 |
| *GABRA2* | 39.51 | 7.54 | 1.07 | 7.07 | 1.54E − 12 | 6.71E − 10 |

Top 50 most differentially expressed genes (*n* = 25 upregulated, *n* = 25 downregulated in heavy vs. low immune infiltrate) between tumor sub-regions based on extent of immune infiltrate.

all tumor specimens. These data are at least consistent with a common neoepitope driving a persistent T-cell response over time. To validate the potential in vitro immunogenicity of the ZDHHC17 p.H507Y neoantigen, we synthesized 12 overlapping candidate 9-mer peptides spanning the point mutation and used these peptides to elicit CD8 T-cell responses from HLA-A*0301 donor peripheral blood mononuclear cells (PBMCs) in peptide stimulation assays in vitro (see Methods). Compared with donor PBMC co-cultured with non-peptide-pulsed A3-K562 cells, we observed elevated IFN-γ production by CD8 T cells of two HLA-A*0301 donors with several peptides (4, 6, 7, 9, 11, and 12) but most particularly from peptides 4 and 12, which induced the most

robust responses at an average of three- to fivefold greater than background levels (i.e., unpulsed cells), thus representing immunogenic epitope candidates (Supplementary Fig. 7A-B).

To evaluate the functional characteristics of this remarkably persistent T-cell clonotype, we harnessed matched single cell *TCRα*, *TCRβ*, and transcriptome sequencing of sorted CD45+ CD3+ T cells within the post-PD-1 inhibitor tumor. The T cells clustered broadly into a population of activated cytolytic T lymphocytes (49%, CTL) expressing *CD8A*, *GZMA*, and *PRF1* and checkpoint-regulated T cells (20%) expressing multiple immune checkpoints including *ICOS*, *CTLA-4*, and *TNFRSF18* but which were also dominantly *CD4* expressing (Fig. 4g, h and

**Table 2 GO term enrichment reveals prominent T- and B-lymphocyte activation in immune-infiltrated tumor samples.**

| GO term | Description | Gene ratio | Background ratio | *p*-Value | *p*-adjust |
|---|---|---|---|---|---|
| GO:0030098 | Lymphocyte differentiation | 36/376 | 344/18493 | 7.01E − 16 | 2.63E − 12 |
| GO:0070661 | Leukocyte proliferation | 31/376 | 283/18493 | 2.26E − 14 | 4.25E − 11 |
| GO:0002429 | Immune response-activating cell surface receptor signaling pathway | 37/376 | 414/18493 | 4.14E − 14 | 5.18E − 11 |
| GO:0002768 | Immune response-regulating cell surface receptor signaling pathway | 38/376 | 445/18493 | 7.79E − 14 | 6.85E − 11 |
| GO:0050851 | Antigen receptor-mediated signaling pathway | 29/376 | 259/18493 | 9.12E − 14 | 6.85E − 11 |
| GO:0042113 | B-cell activation | 31/376 | 303/18493 | 1.46E − 13 | 9.14E − 11 |
| GO:0050854 | Regulation of antigen receptor-mediated signaling pathway | 15/376 | 57/18493 | 3.23E − 13 | 1.73E − 10 |
| GO:0046651 | Lymphocyte proliferation | 28/376 | 264/18493 | 9.51E − 13 | 4.47E − 10 |
| GO:0032943 | Mononuclear cell proliferation | 28/376 | 266/18493 | 1.14E − 12 | 4.78E − 10 |
| GO:0042100 | B-cell proliferation | 17/376 | 91/18493 | 3.53E − 12 | 1.33E − 09 |
| GO:0042110 | T-cell activation | 35/376 | 451/18493 | 1.22E − 11 | 4.16E − 09 |
| GO:0045785 | Positive regulation of cell adhesion | 32/376 | 397/18493 | 3.63E − 11 | 1.14E − 08 |
| GO:1903039 | Positive regulation of leukocyte cell–cell adhesion | 23/376 | 214/18493 | 8.22E − 11 | 2.38E − 08 |
| GO:0007159 | Leukocyte cell − cell adhesion | 28/376 | 327/18493 | 1.59E − 10 | 4.13E − 08 |
| GO:0050870 | Positive regulation of T-cell activation | 22/376 | 202/18493 | 1.65E − 10 | 4.13E − 08 |
| GO:0050852 | T-cell receptor signaling pathway | 19/376 | 150/18493 | 2.23E − 10 | 5.24E − 08 |
| GO:0050867 | Positive regulation of cell activation | 30/376 | 384/18493 | 3.25E − 10 | 7.18E − 08 |
| GO:0022409 | Positive regulation of cell – cell adhesion | 24/376 | 251/18493 | 3.62E − 10 | 7.57E − 08 |
| GO:0050855 | Regulation of B-cell receptor signaling pathway | 9/376 | 24/18493 | 5.39E − 10 | 1.07E − 07 |
| GO:0030183 | B-cell differentiation | 17/376 | 125/18493 | 6.55E − 10 | 1.22E − 07 |

Top 20 most-enriched GO-BP terms within differentially expressed genes between samples displaying high vs. low levels of leukocytic infiltrate.

Supplementary Fig. 7C). We recovered 11 counts of the persistent TCR-Vβ rearrangement in a population of 6267 T cells (0.17%) and identified multiple *TCRα* (*TRAV35-TRAJ23* and *TRAV2-TRAJ2*) and *TCRβ* (*TRBV29-TRBJ2*, *TRBV7-TRBJ2*, and *TRBV5-TRBJ1*) partners to the TCR-Vβ sequence of interest comprising this T-cell population, including T cells with dual TCR-Vβ rearrangements. Based on VDJ combinatorics, a minimum of two distinct T-cell clones contributed to this recurrent TCR rearrangement at the amino acid level. Interestingly, when immunoprofiling these cells using matched RNA sequencing (RNA-seq) data, we found eight cells within the cluster expressing multiple immune checkpoint molecules and five cells in the cluster resembling activated CTLs. The detection of multiple clones at nucleotide level expressing a synonymous CDR3 amino acid sequence, their persistence over nearly a decade, and simultaneous presence of both activated and exhausted phenotypes suggests that this T-cell population arose from multiple independent T-cell priming events rather than functional divergence following a single more recent priming/activation event.

**Chromosome 7 gain is associated with an unfavorable immune outcome**. To further explore the link between genomic CNAs and immune ITH, we focused on the observation of decreased ESTIMATE immune scores in regions with subclonal gain of chromosome 7 ($p = 0.018$, two-sided *t*-test). Immune deconvolution revealed low counts of multiple immune cell subsets including T cells ($p = 0.00096$), CD8+ T cells ($p = 0.084$), cytotoxic lymphocytes ($p = 0.036$), natural killer cells ($p = 0.0013$), B cells ($p = 0.015$), monocytic lineage ($p = 7.6e − 5$), myeloid-derived dendritic cells ($p = 0.0039$), and most significantly neutrophils ($p = 3.4e − 5$; all two-sided *t*-test comparison of means) in these sub-regions (Supplementary Fig. 8A; neutrophil signature gene sets, Supplementary Data 7)[14]. However, although overall neutrophil counts (derived from transcriptome data) were low and this was consistent with generally low neutrophil densities identified by CD15 immunostaining of corresponding FFPE

sections, interrogation of neutrophil activation gene sets (Supplementary Data 8) to assess putative neutrophil functional status revealed a net neutrophil activation (slightly higher levels of "positive neutrophil activation," $p = 0.51$; significantly lower levels of "negative neutrophil activation," $p = 0.00087$, and "negative regulation of neutrophil degranulation," $p = 0.0068$; all Benjamini–Hochberg corrected *p*-values) in the sub-regions with gain of chromosome 7 compared with chromosome 7 stable regions (Fig. 5a, b). We sought to validate this relationship using TCGA melanoma (skin cutaneous melanoma; SKCM) samples and identified samples with both copy number and mRNA expression data ($n = 470$), within which 50 samples harbored whole-chromosome gains of chromosome 7. Differential expression analysis comparing samples with chromosome 7 gain vs. non-gain revealed a marked enrichment for neutrophil-related genes and associated pathway level enrichment (Fig. 5c, d) despite marginally lower neutrophil enumeration by CIBERSORT (Supplementary Fig. 8B): the top four enriched GO terms were neutrophil degranulation, neutrophil involved in immune response, neutrophil activation and neutrophil-mediated immunity (all $p = 1e − 6$, two-sided *t*-test comparison of means with Benjamini–Hochberg correction). In parallel, Kyoto Encyclopedia of Genes and Genomes (KEGG) pathways enriched in the chromosome 7 gain samples included response to bacterial infections, phagosome and lysosome formation, and antigen processing, consistent with the observed strength of gene enrichments in neutrophil-related GO terms (Supplementary Fig. 8C).

As immune infiltrate correlates with overall survival and has been shown to correlate with responsiveness to anti-PD-1 and anti-CTLA-4 immunotherapy, we then investigated the significance of such neutrophil signatures in three publicly available immunotherapy-treated melanoma cohorts ($n = 119$)[5,19,26,27]. Within the anti-CTLA-4 cohort (Van Allen, $n = 36$) and two anti-PD-1 cohorts (Hugo, $n = 27$; Riaz, $n = 56$), overall neutrophil estimation was again largely similar (Supplementary Fig. 8D-F); however, the genes significantly enriched in non-responders

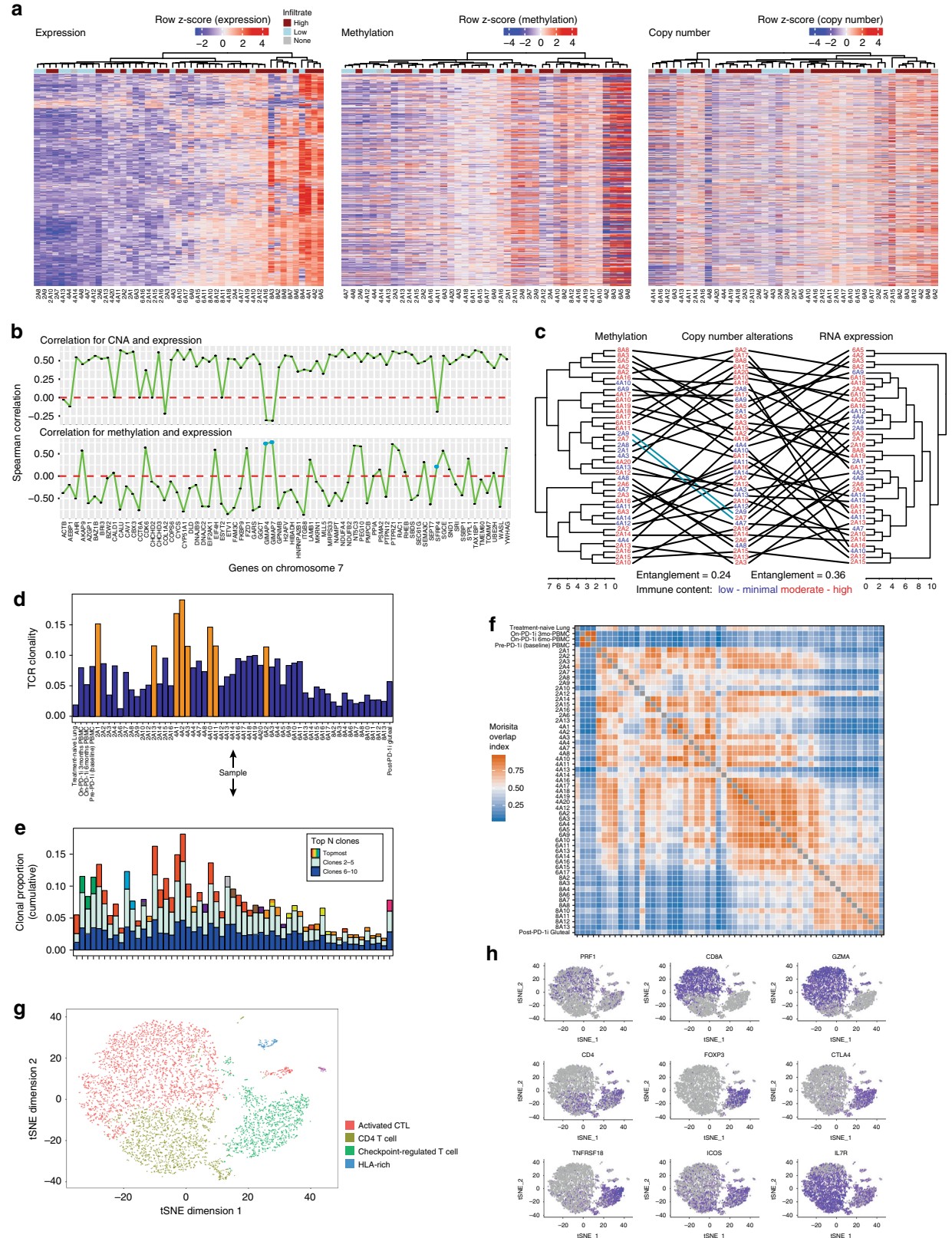

compared with responders to therapy showed pathway level enrichments dominated by the same neutrophil signatures observed in chromosome 7 gain TCGA melanoma samples, namely neutrophil degranulation ($p = 6e − 5$, $3e − 10$, $5e − 6$, respectively), neutrophil involved in immune response ($p = 6e − 5$, $3e − 10$, $5e − 6$, respectively), neutrophil activation ($p = 6e − 5$, $3e − 10$, $5e − 6$, respectively), and neutrophil-mediated immunity ($p = 6e − 5$, $3e − 10$, $5e − 6$ respectively; all Benjamini–Hochberg-corrected $p$-values) (Fig. 5e–g). A core group of differentially expressed genes (*FTH1*, *FTL*, *HSPA8*, *HSP90AA1*, and *HSP90B1*) was recurrently identified within significantly enriched pathways across TCGA melanoma samples and clinical cohorts (Fig. 5h).

**Fig. 4 T-cell repertoire dynamics reveal high ITH, potential for long-term clonal persistence and irregular intratumoral movement. a** Heatmaps of the 560 most variably expressed genes (based on median absolute deviation) across the transcriptome dataset for which matched methylation and copy number data were available, indicating distinct sample-wise clustering patterns within each dataset and generally unidirectional methylation patterns within samples. Data are log2-transformed counts (gene expression), $\beta$-values (methylation), and log2(probe intensity = observed intensity/reference intensity) (copy number), $z$-scored within each data type. **b** Correlation of copy number (upper panel) and methylation (lower panel) with transcript expression for the most variable genes on chromosome 7 showing mostly positive correlations for CNA and mostly negative correlations for methylation as expected. Three genes (indicated in blue, lower panel) showed discordant correlations for both CNA and methylation. **c** Tanglegram showing relationships between the sample clustering obtained independently from each of the copy number, methylation, and transcriptome datasets for samples represented in all datasets. Entaglement values (range 0–1) indicate moderate lack of cluster structure concordance, indicative in this context of significant additional (unmeasured) factors contributing to the regulation of mRNA expression. Immune status of each subregion sample is indicated in color. **d** T-cell receptor-V$\beta$ (TCR) clonality (range 0–1) varied considerably between clinically relevant time points, between tumors and the peripheral blood, and spatially within the on-PD-1 inhibitor tumor. Samples with clonality > 0.1 are highlighted in orange. **e** Top ten most abundant TCR clonotype proportions (i.e., fraction of total identified TCR clonotypes) in each sample are represented as stacked bar plots. The topmost abundant clone is colored at the top of the bar, with each color representing a unique clonotype that may be shared between samples. Clonotypes 2–5 and 6–10 are colored in light blue and deep blue, respectively. **f** Morisita overlap index (MOI, range 0–1) values of TCR repertoires comparing the pretreatment sample, peripheral blood samples, on-PD-1 inhibitor sample (each subregion), and a post-PD-1 inhibitor sample were used to compare the overlap in shared nucleotypes in the TCR repertoire identified in each sample. Higher MOI indicates a greater proportion of shared TCR sequences. Within the on-PD-1 inhibitor sample, TCR clonotypes present in section 8 were largely restricted to this geographic location, which was notably highly necrotic. There was considerable sharing of clonotypes between sections 4 and 6, and to a lesser degree between 2, 4, and 6, suggesting a greater degree of physical movement of T cells between these sections. **g** tSNE plot of TIL populations. The majority of the cells fell into an activated cytotoxic T-cell lymphocyte, CD4 T cell, and checkpoint-inhibited T-cell phenotype. **h** Marker gene expression levels across TIL clusters. Relative expression of key marker genes associated with a cytotoxic T-cell phenotype (CD8, GZMA, and PRF1), CD4 phenotype (CD4 and IL7R), and a multiply checkpoint-inhibited phenotype (FOXP3, CTLA-4, GITR, and ICOS) are overlaid on the tSNE clusters.

Together, these data suggest a recurrent immunosuppressive role of chromosome 7 gain, potentially mediated by neutrophil accumulation and/or activation, although it is unclear whether neutrophil density or activation status are acting as a surrogate for the typical co-localization of tumor necrosis observed at such sites. However, these associations appear active both locally within tumors and at the bulk tumor level where it has clinical implications for immune checkpoint blockade.

## Discussion

In this study, we performed matched genomic and immune analysis of 67 distinct regions of a melanoma metastasis coupled to longitudinal analyses in a patient treated with multiple therapies, including prolonged exposure to (and progression on) anti-PD-1 immunotherapy. Consistent with previous studies, we observed minimal ITH in oncogenic mutations in canonical melanoma driver genes, but reveal striking genomic ITH in CNAs, including distinct copy number loss in chromosome 10 and gains of chromosomes 7 and 13, which may contribute to differences in the immune landscape. The loss of chromosome 10, and thus *PTEN*, has been implicated in resistance to PD-1 blockade previously, and in the context of this immunotherapy-treated patient was observed to be lost in a stepwise manner between tumors sampled prior to, during, and after anti-PD-1 therapy[3,28].

The most immediately apparent implication of the extent of heterogeneity observed and its diverse representation across space even within a single metastatic deposit is how inherently limited the prediction of clinical outcomes can be when based on limited physical sampling of tumor material, especially if only one metastatic deposit is sampled. Indeed, based on transcriptional heterogeneity alone, a complete understanding of the immunogenomic TME of the extensively profiled lesion in this study would likely require a minimum of four separate passes if subjected to needle biopsy. Although the degree of immunogenomic spatial heterogeneity in any given tumor mass cannot yet be predicted non-invasively, the spontaneous nature of immune–tumor interactions implies that relevant spatial heterogeneity will be found, irrespective of prior therapeutic exposures. To the extent that additional non-mutational features

begin to emerge as clinically meaningful biomarkers for treatment response/resistance, these facets of multidimensional heterogeneity need to be considered when planning biopsy-derived, biomarker-driven trials.

Tumor heterogeneity has been linked to the emergence of treatment-resistant tumor cell sub-populations, which expand under the selective pressure of therapy. Conceptually, heterogeneity encompasses multiple domains (e.g., spatial, temporal, and clonal) and can be applied to any measurable feature of a tumor; thus, it remains unclear exactly which molecular constituents of heterogeneity are most consequential to clinical outcomes. Previous studies of heterogeneity in other tumor types (e.g., renal, prostate, and lung) have focused primarily or exclusively on phylogenetic mutational analyses to characterize clonal heterogeneity of tumor cell content[29–32]. When performed, multi-region sequencing either for tumor cell mutations or TCR profiling has surveyed minimal numbers of regions (e.g., three to five per tumor), often in relatively small numbers of samples[31,33–35], whereas truly multi-platform analyses have effectively evaluated inter-tumoral rather than ITH[36]. At the extreme of cellular resolution, studies employing single cell techniques, whilst informative of the multidimensional cellular heterogeneity within bulk tumor cell populations, necessarily destroy spatial information during sample processing and arguably do not comprehensively survey the transcriptome within any individual cell[11]. Thrane and colleagues performed a proof-of-principle high resolution spatial transcriptomics analysis of four lymph node metastases obtained from patients with stage III melanoma, finding evidence of variably distinct gene expression profiles between regions of tumor, lymphoid tissue, and an apparent transition zone that may have represented functional interaction between tumor, stroma and lymphoid cells[37]. Relative intratumoral transcriptomic homogeneity in one sample was associated with long-term overall survival, however other domains of heterogeneity were not evaluable with this technique.

We found chromosome 7 gain to be significantly associated with features of an unfavorable immune microenvironment, including a paucity of effector cell populations and signatures of neutrophil activation. This relationship was confirmed among melanoma samples of TCGA. Furthermore, a strikingly consistent

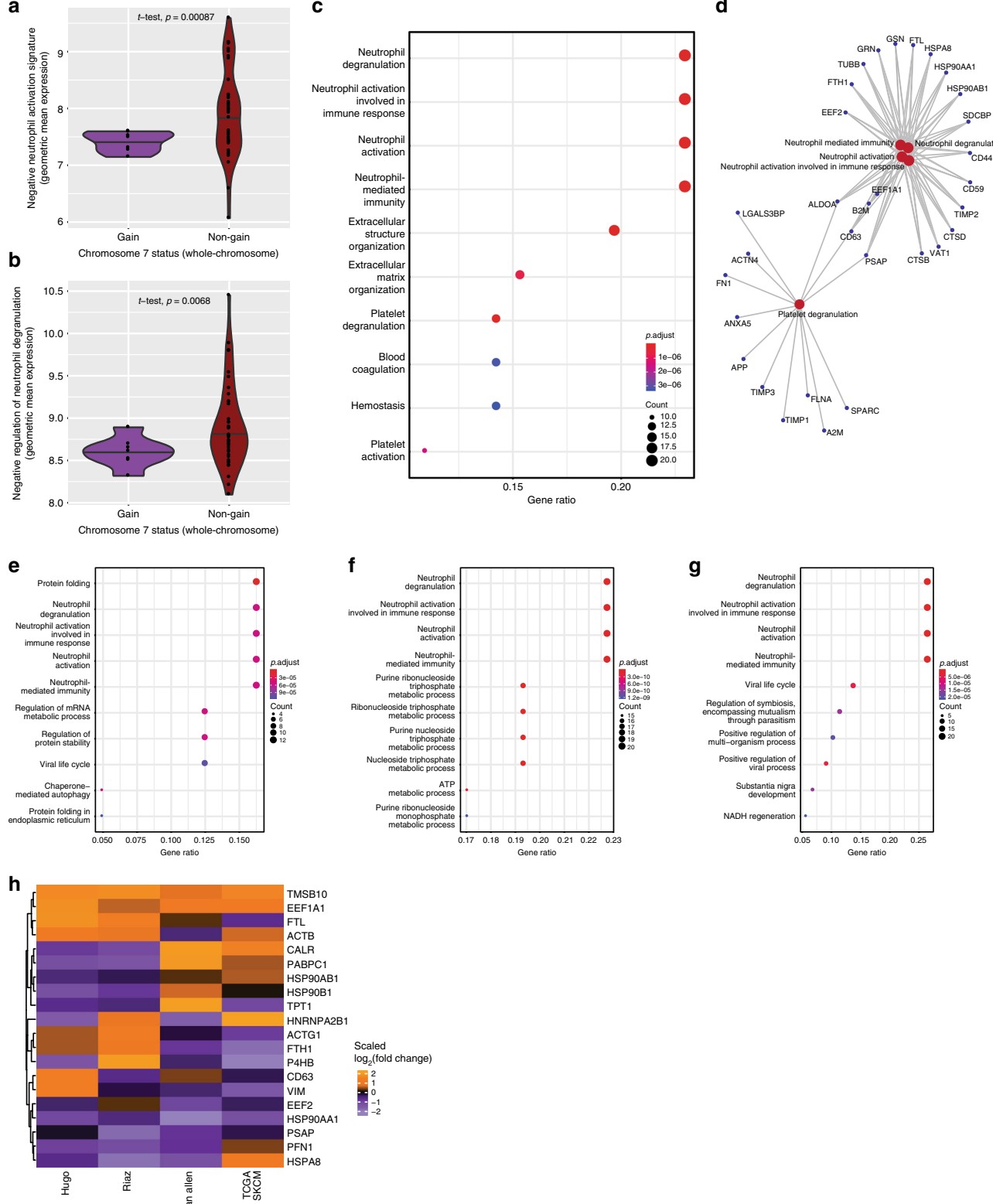

set of neutrophil enrichments was observed in melanoma tumors failing to respond to either anti-CTLA-4 or anti-PD-1 therapy across three independent published cohorts. This reveals two key messages, with the important caveat that additional studies are required to clarify the nature of neutrophil recruitment and activation in anti-tumor-immune responses and whether their presence is largely as a consequence of cellular destruction by other mechanisms. First, the insights from regional immunogenomic differences within a single tumor metastasis can directly translate to the bulk tumor level and, second, chromosome 7 gain may drive an immunologically adverse phenotype associated directly or indirectly with neutrophil activation. Several prominent oncogenes (*BRAF*, *EGFR*, and *MET*) are located on chromosome 7 and may thus be subject to amplification in the setting

**Fig. 5 Chromosome 7 gain is associated with an unfavorable immune environment driven by neutrophil signatures that characterize non-responders to immune checkpoint blockade. a, b** Chromosome 7 gain sub-regions of the melanoma mass progressing during anti-PD-1 therapy revealed lesser suppressive neutrophil signatures compared with sub-regions unaffected by chromosome 7 gains. Scores shown represent geometric mean log2-transformed counts of genes within the GO terms indicated (see also Supplementary Data 7). Plots include two-sample $t$-test comparison with accompanying significance values. **c** Samples affected by whole-chromosome 7 gains within The Cancer Genome Atlas skin cutaneous melanoma (SKCM) dataset revealed prominent differential upregulation of genes involved in neutrophil activation signatures revealed by Gene Ontology term enrichments. Gene ratios indicate the ratio of representation of input genes within the indicated GO term gene set relative to all queried GO term gene sets. Top enriched pathways are displayed after Benjamini–Hochberg correction for multiple testing. **d** Connected gene network of genes involved in major GO term enrichments within chromosome 7 gain TCGA SKCM samples. **e–g** Differentially expressed genes contrasting responders vs. non-responders to immune checkpoint blockade agents in publicly available datasets revealed consistent GO term enrichments for neutrophil activation signatures. Top enriched pathways are displayed after Benjamini–Hochberg correction for multiple testing. **e** Van Allen et al.[5], anti-CTLA-4 dataset; **f** Hugo et al.[27], anti-PD-1 dataset; **g** Riaz et al.[26], anti-PD-1 dataset. **h** Heatmap of genes found recurrently enriched across TCGA SKCM chromosome 7 gain and clinical non-responder samples showing relative enrichment of each gene across the datasets as a scaled value.

of copy number gains. In our patient's spatially profiled tumor, *BRAF* amplification compounded by an activating *BRAF*[G421R] mutation may have contributed to immunosuppressive MAPK signaling, potentially in conjunction with other chromosome 7 oncogenes[24,38]. We also observed recurrent dysregulation of ferritin and HSP90-related genes, suggestive of an enhanced acute-phase protein reaction, iron loading and molecular stress in the context of chromosome 7 gain and immunotherapy failure. Iron availability is known to influence tumor cell survival and the function of numerous immune cell types including T cells; however, these competing outcomes have been poorly studied in solid tumors such as melanoma[39,40]. Nevertheless, a potential role for immunosuppressive neutrophil phenotypes and iron trafficking within the TME warrants further evaluation.

We identified a persistent, high frequency T-cell clonotype prevalent in multiple tumor deposits over many years in this patient, with evidence of both activated and checkpoint molecule regulated (likely previously activated) cells present simultaneously. The time frame, and multiplicity of independent genomic rearrangements leading to this clonotype indicates repeated priming events, potentially in response to a highly persistent tumor antigen robust to multiple lines of treatment. Identification of such a persistent T-cell population, or its persistent antigen, might be specifically useful for the development of defined antigen therapies such as vaccines (definitive or adjunctive therapy) or engineered T cell therapies based upon these targets, and warrants wider sampling of multiple tumors in patients—including the use of archival tissues—to identify persistent tumor features that may be exploited for therapeutic advantage. Furthermore, our integrative immunogenomic analysis strongly suggests that high frequency T-cell clonotypes may be recruited to the tumor microenvironment not only due to tumor cell reactivity, but as passengers in the inflammatory milieu. Further work will be required to determine to what extent such "passenger" T-cell clonotypes contribute usefully to the anti-cancer immune response.

Our findings of extensive immunogenomic heterogeneity at the intratumoral level are inherently limited by detailed multi-platform profiling of a single lesion; thus, it is difficult to determine how typical the observed extent of heterogeneity is to broader patient populations, particularly those having differing clinical scenarios and treatment outcomes. Nonetheless, considering that subclonal variation has now been described in numerous tumor types, these findings serve to highlight the potential sensitivity of the immune microenvironment to local factors, including tumor genomic features that appear to have functional impact on local tumor immunity. Through analyses of several clinical datasets we found certain immunogenomic features from our deeply profiled tumor to have meaningful correlates in additional cohorts of patient samples, but additional

studies are clearly required to refine these inferences towards therapeutically manipulable strategies. Whilst overall objective clinical responses were not achieved in this patient over a period approaching 10 years and 7 lines of therapy, substantial clinical benefit was derived. Although this patient represents only a subset of long-term survivors with metastatic melanoma, considering the increasing availability of disease modifying therapies, it is likely that this group will become increasingly prevalent. Further molecular characterization will ultimately aid in understanding long-term survivors of metastatic disease, providing therapeutic insights transferrable to the greater majority of patients.

## Methods

**Biospecimen collection.** Patient data, tumor samples, and matched peripheral blood leukocyte samples were obtained and used in accordance with research protocols approved by the local Institutional Review Board of the University of Texas MD Anderson Cancer Center. Biospecimens were retrieved, collected, and analyzed under UT MD Anderson Cancer Center Institutional Review Board-approved protocols in accordance with the Declaration of Helsinki. In particular, the patient provided informed consent to approved institutional protocols that cover specimen collection and storage, and the use for research purposes, along with collection and consent to publish relevant de-identified clinical metadata pertinent to this study.

**Sample processing: spatial intratumoral analysis.** We developed a three-dimensional model for processing a whole resected metastatic lesion. The lesion measured 2.5 cm × 2.4 cm × 1.5 cm and was obtained from abdominal wall soft tissue.

Processing consisted of the following steps:

1) Following resection, the specimen was measured and oriented according to its largest diameter. Lateral (short axis, "left"/"right") orientation was preserved by differential inking of the outside surface with red or blue ink (Fig. 1b).
2) The specimen was serially sectioned perpendicularly to its largest axis resulting in eight slices of 2–3 mm thickness. The cut surface was then painted prior to each cut to preserve the true orientation.
3) Alternate slices were submitted for FFPE (four slices; odd-numbered slices) or frozen (four slices; even-numbered slices) processing. FFPE slices were used for pathological assessment and IHC analysis. Frozen slices were embedded in optimal cutting temperature compound and used for DNA, RNA, and protein extraction and downstream analyses.
4) For frozen sections, hematoxylin and eosin staining was performed for histological quality control (QC). Frozen sections were further squared into a 0.2–0.4 cm grid as shown in Fig. 1b and Extended Data Fig. 3, generating a total of 67 sub-regions of tumor. Due to the variation in tumor shape throughout three-dimensional space, each tumor slice presented a distinct cross-sectional area and thus a unique grid was applied to each frozen tumor slice (slice $n$) and the immediately adjacent FFPE slices (slices $n − 1$ face B and $n + 1$ face A). Thus, although subregion numbering generally proceeded bottom-to-top and right-to-left, specific slice subregion numbering is not directly comparable between slices. Each piece was labeled and numbered.
5) Histopathological review for assessment of viable tumor, inflammatory infiltrate, necrosis and connective tissue of each subregion piece was performed by a pathologist.

   1) Designation of sub-regions as located at the tumor core or margin was performed by inspection of all regions annotated on SOX10-stained IHC

slides to infer a volumetric estimate of tumor content and location. Three-dimensional variation throughout the frozen slices was accounted for by considering the immediately adjacent FFPE slices (ie: above and below, when both were available) in order to arrive at a consensus call to categorize tumors as core or margin.

2) Immune infiltrates were evaluated using CD45-LCA positive cell density measured by digital image quantification using the Aperio ImageScope software. CD45+ density was then categorized as low, medium, or high by binning into lower, middle, or upper tertiles considering all regions analyzed. Several additional factors required consideration before arriving at a final semi-quantitative categorization as having focal, low, moderate, or high immune infiltration: between-slide variation in staining efficiency, spatial distribution of immune cell content (focal, broad, intra/peri-/extra-tumoral), and a consensus estimate for genomic frozen slices by considering the adjacent FFPE sections on both sides for which the higher immune content was assigned precedence.

6) Frozen-section squares were submitted for dual DNA and RNA extraction, and for protein extraction.

7) DNA samples were submitted for T200 targeted sequencing ($n = 38$), methylation ($n = 38$), whole exome sequencing ($n = 6$), and TCR sequencing ($n = 46$). RNA samples were submitted for gene expression profiling by RNA sequencing ($n = 48$) and TCR sequencing ($n = 46$).

**Sample processing: longitudinal time points**. Archival formalin-fixed paraffin-embedded specimens from the lung metastasis (pretreatment) and right gluteal mass (post-PD-1 inhibitor) were obtained from the institutional pathology department and utilized for tumor evaluation by a pathologist, DNA/RNA extraction, and IHC as described below.

A single-cell suspension was generated from the post-PD-1 inhibitor time point tumor by gentle mechanical digestion of fresh tumor material, followed by enzymatic digestion with 2 mg/mL collagenase A (Roche, catalog number 11 088 793 001) and DNase I (Roche, catalog number 11 284 932 001) in serum-free RPMI 1640 (Gibco, catalog number 11875119) for 1 h at 37 °C with agitation. The cryopreserved single-cell suspension was thawed and purified for viable cells by negative selection using the MACS Dead Cell Removal Kit (catalog number 130-090-101) and an LS Column (catalog number 130-042-401) on the QuadroMACS Separator (catalog number 130-090-976, all Miltenyi Biotec). The purified single-cell suspension was stained with SYTOX blue dead cell stain (catalog number S34857, Thermo fisher Scientific), anti-human CD45 PerCP-Cy5.5 (clone HI30, catalog number 564105, BD Biosciences), anti-human CD3 FITC (clone SK7, catalog number 340542, BD Biosciences), and anti-human melanoma (MCSP) APC (catalog number 130-091-252, Miltenyi Biotec) prior to cell sorting on a BD FACSAria III flow cytometer (BD Biosciences) to enrich for a live T-cell fraction (CD45+ CD3+) and live tumor fraction (CD45− MCSP+)[41].

**Nucleic acid extraction**. DNA and RNA isolation were performed using the AllPrep DNA/RNA/miRNA Universal kit (catalog number 80224, Qiagen) for fresh-frozen samples and the AllPrep DNA/RNA FFPE kit (catalog number 80234, Qiagen) for FFPE samples according to the manufacturer's instructions. Tumor viability of 80% estimated from corresponding IHC samples was set as a minimum threshold for genomic analyses. Samples with DNA integrity numbers > 7 were used for targeted panel sequencing and EPIC array methylation profiling. RNA-seq was performed on samples with a minimum RNA integrity number (RIN) of 5.5, except for two cases (6A10 and 8A3) with RINs > 3. A minimum of 700 ng of RNA were required for all samples undergoing RNA-seq.

**Cancer gene panel DNA sequencing**. Samples with cancer cell purity greater than 80% based on pathologic assessment were used for cancer gene panel DNA sequencing. Mean sequencing coverage was 861× in tumors and 1314× in germline samples. Paired-end reads in FASTQ format were generated by the Illumina pipeline and aligned to the reference human genome hg19 build using the Burrows-Wheeler Alignment Tool (BWA, v0.7.5) with default settings[42]. Aligned reads were further processed using GATK with best practices for removing duplicates, indel removal, and recalibration[43].

To detect potential single-nucleotide variants, MuTect (v1.1.4)[44] was used with default parameters including a VAF of >10% in tumor DNA, variants present on both strands, a high read count of tumor DNA and the removal of positions listed in dbSNP 129. Pindel (v0.2.4) was used to identify small insertions and deletions[45]. Copy number was called using Sequenza (v2.1.2)[46]. Tumor purities and ploidies were calculated from Sequenza calls using the sequencing data with default parameters. The content of the cancer gene panel is given in Supplementary Data 2.

**Whole exome sequencing**. Exome sequencing data were generated using methods as previously described, including library preparation using the Agilent SureSelect XT Target Enrichment protocol (#5190-8646) prior to sequencing on an Illumina HiSeq 2000/2500 v3 system using 76 bp paired-end reads[3]. Raw sequencing data were then processed using Saturn V, the next-generation sequencing data processing and analysis pipeline developed by the Department of Genomic Medicine

at the UT MD Anderson Cancer Center. BCL files were pre-processed using CASAVA (Consensus Assessment of Sequence and Variation, v1.8.2) for demultiplexing and converting to FASTQ. The files were aligned using the BWA, v0.7.5 using the hg19 reference genome build[42]. Picard (v2.5.0) was used to convert SAM files to BAM files and remove duplicates. BAM files were realigned and recalibrated using GATK. Mean coverage was 181× for tumors and 81× for matched germline DNA. MuTect and Pindel were used to identify somatic point mutations and small insertions and deletions, respectively[44,45]. Somatic mutations in HLA genes were called using POLYSOLVER (v1.0)[47]. Data were annotated by ANNOVAR (v20180118) using the NCBI Reference Sequence Database[48].

**Phylogenetic tree construction**. Mutations that passed our WES Mutect filtering criteria were considered for the purpose of constructing phylogenetic trees. A tumor power of 0.8 was used to filter mutations that reflected the power to detect a mutation at 0.30 allelic fraction. Trees were built using binary presence/absence matrices built from the distribution of mutations within the tumor samples. A representative subregion sample was chosen from the four frozen slices of the on-PD-1 inhibitor tumor (Fig. 1b), thus producing a total set of six samples being compared (pretreatment ×1, on-PD-1 inhibitor ×4, post-PD-1 inhibitor ×1). As the most inferior section (section 8) of the on-PD-1i tumor was largely necrotic, we sampled two regions from the preceding frozen section (final samples: 2A2, 4A11, 6A3, 6A16). To compare the three time points at bulk tumor level, we combined the multiple on-PD-1 inhibitor tumor sub-samples. The R Bioconductor package phangorn (v2.5.5) was utilized to compute the hamming distance under the neighbor-joining tree method and generated unrooted trees[49]. The distance was computed after 100 bootstrap iterations with the bootstrap value reflected on the branch. We identified somatic mutations using MuTect and both DNA copy number changes and tumor purity using Sequenza[44,46]. We estimated the cancer cell fractions identified with a particular mutation, accounting for tumor purity, using PyClone (v0.13.0)[50], which was used to infer cancer cell fractions and assign clonal clusters[50]. Further clonal evolution was evaluated through ClonEvol (v0.1)[51].

**RNA sequencing**. Paired-end transcriptome reads were aligned using TopHat2, to the UCSC hg19 reference genome[52]. Gene read counts were generated using Htseq-count[53]. Bioconductor R package DESeq2 (v1.24.0) was used to normalize the read counts and for downstream analysis, vsn (v3.52.0) was used for variance stabilization[54,55]. Differential gene expression analysis between the heavy and low infiltrated samples were performed after adjusting for variation in tumor content due to core or margin location by excluding samples with very low tumor purity (i.e., sampling predominantly surrounding stroma). Genes with significant changes in expression were assessed by including the top 1000 most variant genes after performing median absolute deviation. The genes were clustered based on Euclidean distance and the samples based on Pearson correlation with complete linkage. DAVID (v6.8) online functional annotation tools, showed immune-regulated pathways from the most variable genes with an FDR cutoff of 1%[56]. Pathway analysis was performed on the most differentially expressed genes. ssGSEA was run through GSVA (v1.32.0)[57]. Pathway level enrichment was run on the output of DESeq2 for each condition through DOSE (v3.10.2) and ClusterProfiler (v3.12.0)[54,58–60]. Cell type-specific gene expression was evaluated using immune and melanoma-specific markers. All heatmaps were constructed using Complex-Heatmap (v2.0.0)[61]. ESTIMATE (v1.0.13) was used to detect tumor purity and the presence of infiltrating stromal/immune cells in tumor tissues using gene expression data[12]. Sample distances were visualized using Circlize (v0.4.8) and Plotly (v4.9.0)[62,63]. Consensus clustering was performed using the Consensus Cluster Plus (v1.48.0)[64]. Hierarchical clustering was used to group the 48 transcriptomic samples with a maximum cluster count of 20. The delta area plot was used to determine the relative increase in consensus clustering of samples within a given cluster and to determine a value of $k$ beyond which no appreciable increase was achieved. The tracking plot was used to depicts which samples was allocated to which cluster, and lack of correlation with geographic location of the sample. For validation across public datasets, we used reads/fragments per kilobase of transcript per million mapped reads (FPKM) values to build linear models of expression, given the design matrix of binary responders and non-responders using response classifications based on RECIST v1.1 criteria provided with each paper. To compare the most representative samples across all datasets, we restricted our analysis from the Riaz dataset to include only the on-treatment time point samples. Ribosomal L and S (RPL and RPS, respectively) genes were excluded from downstream analysis, following which the expression values were log-transformed. Linear modeling was performed to fit a model of expression values for each gene, given the binary response status[65]. Empirical Bayes moderation was then carried out by utilizing information across all genes to obtain precise estimates of gene-wise variability[66]. The differentially expressed genes were characterized by FDR-adjusted p-values of <0.05 and log fold change in the positive direction for the non-responders. Downstream GO and KEGG pathway enrichment was performed using ClusterProfiler[59] based on a logFC > 7 and an adjusted P-value of <1e − 5.

**Methylation analysis**. We studied methylation levels through the pipeline integrated into the Bioconductor package, ChAMP (v2.14.0) using R[67]. The data were imported as raw idat files and a variety of QC plots were evaluated. The data was

imported and filtered based on detection $p$-value (<0.01) and probes with <3 beads in 5% of samples per probes. Non-CPG probes and single-nucleotide polymorphism-related probes were then removed. Finally, multi-hit probes and probes located on the X chromosome were filtered out.

Type II probe normalization was performed using BMIQ (v1.5)[68]. We also assessed the number and nature of significant components of variation by using singular value decomposition to look at batch effects and COMBAT (v3.32.1) for batch correction[69]. For the identification of differentially methylated regions, the BumpHunter (v1.26.0) method was used to identify extended segments of the genome that show quantitative alteration in DNA methylation levels[70].

We used the mean normalized value of beta to collapse probe level data to gene-wise data for integrative analysis. Genes on chromosome 7 were used to compare the correlation of expression and methylation. The gene level beta values for the most variable genes from expression were used to compare with copy number and expression data.

**TCR sequencing.** DNA-based: DNA sequencing of the variable region of the β-chain of the TCR was performed by ImmunoSeq (Adaptive Biotechnologies, Seattle, WA)[71,72]. RNA-based: RNA-seq of the variable chain of the TCR was performed using Immunoverse TCR (ArcherDX, Boulder, CO)[73]. The TCR clonality was used for linear regression. The top 5 clones with respect to DNA and RNA clonal fractions were calculated for each sample, and their residuals from the line of best fit were used as a measure of activation of TCR clones. Spearman's rank correlation was performed between the residuals and the clonality measured from DNA-based TCR sequencing. The top clones were analyzed using both platforms independently and then concurrently. Plot3D was used to map the potential trajectory of the most dominant clones across all regions[74]. TCR statistics were computed using the R package tcR (v2.2.4)[75].

**Stimulation of ZDNNC17 p.H507Y neoantigen-specific T cells.** To evaluate the potential in vitro immunogenicity of the ZDNNC17 p.H507Y neoantigen, we synthesized 12 overlapping candidate 9-mer peptides (ELIM Biopharmaceuticals, Inc., Hayward, CA), spanning the neoantigenic point mutation, and used these peptides to elicit T cell responses from HLA-A*0301 donor PBMC. Peptides were dissolved in 1× phosphate-buffered saline at a concentration of 10 mg/mL. HLA-A*0301-transfected K562 (A3-K562) cells pulsed with 2 µg/mL of each ZDNNC17 p.H507Y peptide were used as antigen-presenting cells for stimulating CD8+ T cells from each of two HLA-A*0301 donors using methods previously established in our lab[76]. For each peptide stimulation, irradiated (8000 rads) peptide-pulsed K562 cells were co-cultured in a 48 well plate with 1 million PBMC from each donor at a ratio of 1:20 in RPMI 1640 containing 25 mM HEPES, 2 mM L-glutamine, 10% human AB serum (CTL medium), and β2-microglobulin (3 µg/mL)[76]. Three rounds of PBMC stimulation with peptide-pulsed A3-K562 were performed at 7-day intervals. During the first stimulation, IL-21 (30 ng/mL; Peprotech, Rocky Hill, NJ, USA) was included in the cell culture medium and during the second and third stimulation cycle, IL-21 (30 ng/mL), IL-2 (10 ng/mL; Bayer, Terrytown, NY, USA), and IL-7 (5 ng/mL; R&D Systems, Minneapolis, MN, USA) were added to the growth medium as previously described[76,77]. Controls included PBMC co-cultured with non-pulsed A3-K562 cells and A3-K562 pulsed with a pool of all 12 peptides.

After three rounds of stimulation, an aliquot of 100,000 cells from each well was co-cultured overnight with peptide-pulsed K562 cells at a ratio of 10:1 to assay for antigen specific T cells using flow cytometry-based intracellular IFN-γ production assay[78]. The cells were cultured in the presence of the intracellular protein transport inhibitor Brefeldin A (Thermo Fisher, USA). After overnight culture, cells were washed and then stained with CD8-APC (Clone K1, BioLegend, San Diego, CA) for 20 min. Intracellular staining for IFN-γ-PE (Clone B27, BioLegend, San Diego, CA) was performed according to the manufacturer's protocol. After staining, cells were resuspended in 100 µL of fluorescence-activated cell sorting (FACS) buffer and data were acquired using a NovoCyte Flow Cytometer (ACEA Biosciences, San Diego, CA). Data were analyzed using FlowJo™ software (Tree Star, Ashland, OR, USA). The percentage of background IFN-γ-positive cells was determined by the response of PBMC co-cultured with non-pulsed K562 cells and peptide-specific responses were registered as positive if the proportion of T cells producing IFN-γ in response to stimulation with ZDNNC17 p.H507Y-derived peptide was ≥2-fold higher than the background proportion of IFN-γ + CD8 T cells[79]. PMA + ionomycin treatment of PBMC served as a positive control for the IFN-γ production assay.

**Single-cell sequencing.** Three technical replicates of FACS-sorted T cells (CD45+ CD3+) and one replicate of FACS-sorted tumor cells (MCSP+) were loaded to a targeted 10,000 cells per lane on the 10× Genomics Chromium Controller with the single cell 5' Immune Repertoire and Gene Expression profiling kit. In total, we loaded ~ 30,000 individual tumor-infiltrating lymphocytes and ~10,000 melanoma cells on the 10× platform (10× Genomics, CA, USA). Reverse transcription, TCR enrichment, and library preparations were performed according to the 10× Genomics 5' V(D)J protocol revision C. Transcriptome libraries were pooled and sequenced on the Illumina NovaSeq 6000 S2 flow cell with 26 R1, 8 i7, and 91 R2 cycles, respectively. The TCR libraries were pooled and sequenced on the Illumina

MiSeq V2 150 cycles paired-end. Single-cell transcriptomic and TCR data were processed with the 10× Genomics Cell Ranger Pipeline version 2.2.0 with the software-provided GRCh38 reference transcriptomes[80]. After QC, there was RNA-seq profile data available from 6267 immune and 4303 melanoma cells. Downstream processing and visualization was encompassed through Seurat and tSNE plots[81,82].

**Neoantigen prediction.** HLA Class I neoepitopes were predicted for each sample and affinity was predicted for the predicted peptides using NetMHCpan (v2.8)[83]. Patient HLA-A, HLA-B and HLA-C variants were identified using ATHLATES (v2014_04_26)[84]. All possible 9- to 11-mer peptides flanking a nonsynonymous exonic mutation were generated and binding affinity was predicted based on patient HLA and compared with the wild-type (WT) normal peptide counterpart from NetMHCpan[83]. MuTect calls were filtered using tumor count > 30, normal count > 10, tumor VAF > 0.05, normal VAF < 0.01, and tumor power > 0.8[44]. In addition, a FPKM count > 1 and an alternate allele count > 4 was leveraged from the RNA-seq data. Mutated peptides with predicted $IC_{50}$ < 500 nM were considered to be predicted neoantigens.

**Copy number alteration analysis.** Sequenza was used to obtain copy number segments of $\log_2$ copy ratios for tumor samples[46]. CNTools (v1.24.0) was used to identify copy number gain/loss events at $\log_2$ thresholds of 0.6[85]. The burden of copy number gain or loss was extrapolated from the total number of genes with copy number events in each sample. ExomeCNV (v1.4) was used to calculate the $\log_2$ copy ratios[86]. For the TCGA dataset, processed segmented values were used. Whole-chromosome 7 events were characterized as log segmented mean values >0.3 and covering >70% of the length of the chromosome.

**Reverse-phase protein array.** Frozen tumor subregion samples were processed for RPPA analysis in the UT MD Anderson Cancer Center RPPA Core Facility using previously described methods (https://www.mdanderson.org/research/research-resources/core-facilities/functional-proteomics-rppa-core/rppa-process.html). Briefly, tumor lysates were prepared in RPPA lysis buffer, serially diluted and printed onto nitrocellulose-coated slides prior to being probed with ~300 validated primary antibodies and detection with biotinylated secondary antibodies specific for the primary antibody species. Signal amplification and visualization by a 3,3'-diaminobenzidine (DAB) colorimetric reaction was performed prior to slide scanning and quantification using the Array-Pro Analyzer software (MediaCybernetics), relative protein level estimation using SuperCurve GUI, correction for spatial bias and QC check of each slide.

**Ba/F3 transformation assay.** Transforming potential of JAK1 WT and variants were assayed in IL-3-dependent Ba/F3 cell model as described previously[87]. Briefly, lentivirus vector of JAK1 WT and variants were generated with pHAGE-PURO backbone by high-throughput mutagenesis and molecular barcoding technique as described previously[88]. All clones were full-length validated by Sanger sequencing. Virus were produced by transfecting LentiX-293T cells (Clontech) with pHAGE-PURO backbone and two packaging plasmids (psPAX2 and pMD2.G), and were collected by filtration through 0.45 µm polyvinylidene difluoride filter 3 days post transfection. Six hundred thousand Ba/F3 cells were transduced by spinoculation at 1000 g for 3 h in the presence of polybrene (EMD Millipore; final concentration: 8 µg/mL), and then incubated in the assay medium without IL-3 (Advanced RPMI 1640 with 1× GlutaMAX and 5% fetal bovine serum; Thermo Fisher Scientific) for 2 weeks. Cell viability was measured by CellTiter-Glo luminescent cell viability assay (Promega).

**Immunohistochemistry.** IHC was performed on each of the four FFPE sections using an automated stainer (Leica Bond Max, Leica Biosystems) using primary antibodies against SOX10 (polyclonal, 1:50, Cell Marque, catalog number 383A-7), CD45-LCA (clones 2B11 + PD7/26, 1:300, Dako, catalog number M0701), CD45-RO (clone UCHL1, undiluted, Leica Biosystems, catalog number PA0146), CD4 (clone 4B12, 1:80, Leica Biosystems, catalog number NCL-L-CD4-368), CD8α (clone C8/144B, 1:25, ThermoScientific, catalog number MA5-13473), Granzyme B (clone GrB-7, 1:25, ThermoScientific, catalog number MA1-35461), FoxP3 (clone 206D, 1:50, BioLegend, catalog number 320102), LAG-3 (clone D2G4O, 1:100, Cell Signaling Technology, catalog number 15372), PD-1 (clone EPR4877(2), 1:250, Abcam, catalog number ab137132), PD-L1 (clone E1L3N, 1:100, Cell Signaling Technology, catalog number 13684), PAX5 (clone 1EW, undiluted, Leica Biosystems, catalog number PA0552), CD68 (clone PG-M1, 1:450, Dako, catalog number M0876), CD57 (clone HNK1/Leu-7, 1:250, Abcam, catalog number ab187274), and phospho-p44/42(Erk1/2)(Thr202/Tyr204) (clone D13.14.4E, 1:300, Cell Signaling Technology, catalog number 4370). Slides were counter-stained with hematoxylin, scanned using an Aperio slide scanner (Aperio AT Turbo, Leica Biosystems), and digitized images analyzed using the Aperio ImageScope software (Aperio–Leica Biosystems). Three-dimensional reconstruction re-connecting the frozen and FFPE slices in a sequential order was performed based on the documented inter-slice relationships and histological findings. The IHC slices were gridded into smaller pieces in the ImageScope software to match the gridding of frozen sections, and the results were obtained for each subregion.

IHC-derived cell subset results were quantified as the number of positive-staining cells for each antibody per mm², using custom-tuned algorithms based on nuclear v9, membrane v9, or cytoplasmic v1 algorithms as appropriate for the staining pattern of each antibody.

**Data analysis and statistical considerations**. Statistical analyses were performed using R v3.5.0[89]. Analysis packages and tools used are described in the relevant methods sections. Statistical tests included Welch's two-sample *t*-test and Spearman's rank correlation with the Benjamini–Hochberg correction for an adjusted *p*-value threshold of 0.05. The R package plot3D and Plotly were used to map sequencing-derived data to spatial locations[63,74]. Data were parsed and organized through R packages tidyr, reshape2, and dplyr[90–92]. Dendograms and tanglegrams were constructed using dendextend[93]. Plotting was done through ggplot2 and ggrepel[94,95].

**Reporting summary**. Further information on research design is available in the Nature Research Reporting Summary linked to this article.

## Data availability

Sequencing data (whole exome sequencing, RNA sequencing, T200 targeted gene panel sequencing, and single-cell sequencing) have been deposited in the European Genome-Phenome Archive under accession EGAS00001003292. Access is restricted to non-commercial uses for cancer research purposes in accordance with the relevant patient consent(s) and requests for access to these datasets and to TCR sequencing datasets should be made to AFutreal@mdanderson.org. Other transcriptomic datasets analyzed in this study can be retrieved from dbGAP under the accession dbGaP phs000452.v2.p1 for the Van Allen dataset, and from the GEO repository under the accessions GSE78220 for the Hugo dataset and GSE91061 for the Riaz dataset. The TCGA melanoma dataset can be accessed on the GDC portal (portal.gdc.cancer.gov, cohort TCGA SKCM) (https://portal.gdc.cancer.gov/projects/TCGA-SKCM). Remaining data are available in the Article, Supplementary Information files, or available from the authors upon request.

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

## Acknowledgements

This research was supported by the generous philanthropic contributions to The University of Texas MD Anderson Cancer Center Moon Shots Program™ supporting platform assistance from the Cancer Genomics Laboratory. Additional support was provided to PAF from the Cancer Prevention Research Institute of Texas (R1205 01) and Welch Foundation (G-0040). A.M. was supported by the Cancer Prevention and Research Institute of Texas (CPRIT) Research Training Program (RP170067). M.C.A. is supported by a National Health and Medical Research Council of Australia CJ Martin Early Career Fellowship (#1148680). W.R. was supported by the CPRIT Graduate Scholar Award. A.R. is supported by the Kimberley Clark Foundation Award for Scientific Achievement provided by MD Anderson's Odyssey Fellowship Program. Additional support was provided by the NCI Cancer Center Support Grant (P30-CA16672) to the MDACC Flow Cytometry and Cell Sorting Core Laboratory and NCI #CA16672 to the RPPA Core Facility. The results published here are in part based upon data generated by the TCGA Research Network: https://www.canver.gov/tcga.

## Author contributions

Conceptualization: P.A.F., A.J.L., W.J.H., J.A.W. Investigation: A.M., M.C.A., W.R., M.P.D.M., C.W.H., F.C., S.S., A.R., K.N., A.S., D.S.T., M.T.T. Provision/acquisition of data and materials: A.M., M.C.A., W.R., M.P.D.M., C.W.H., F.C., S.S., A.R., K.W., S.T.,

K.N., A.S., D.S.T., E.C., S.M.R., C.S., D.W., L.L., C.G., Z.A.C., E.M.B., P.H., M.D., J.Z., C.B., N.N., J.A.W., C.Y., W.J.H., A.J.L., P.A.F. Formal analysis: A.M., M.C.A., W.R., S.S., F.W., X.M., X.S. Data curation: A.M., M.C.A., W.R., F.W., X.M., X.S., J.Z. Writing: A.M., M.C.A., W.R., P.A.F. Visualization: A.M., M.C.A., C.W.H. Supervision: J.Z., C.B., N.N., P.S., J.P.A., C.Y., A.J.L., P.A.F. Funding acquisition: P.A.F., A.J.L., E.M.B., J.A.W. All authors read and approved the final manuscript.

## Competing interests

The authors declare the following competing interests: M.C.A. reports advisory board participation and honoraria from Merck Sharp and Dohme, outside the submitted work. Z.A.C. is currently an employee of Medimmune. D.W. is currently an employee of Vedanta Biosciences. CNS is currently an employee of Parker Institute for Cancer Immunotherapy. A.J.L. reports personal fees from Merck, Bristol-Myers Squibb, Novartis, and Roche/Genentech; personal fees and non-financial support from ArcherDX and Beta-Cat; grants and non-financial support from Medimmune/AstraZeneca and Sanofi; and grants, personal fees, and non-financial support from Janssen, all outside the submitted work. P.H. reports consultant or advisor fees from Dragonfly Therapeutics, GlaxoSmithKline, Immatics, and Sanofi. P.S. reports consultant or advisor fees from Bristol-Myers Squibb, GlaxoSmithKline, AstraZeneca, Amgen, Jounce, Kite Pharma, Neon, Evelo, EMD Serono, and Astellas; stock from Jounce, Kite Pharma, Evelo, Constellation, Neon; and has a patent licensed to Jounce, all outside the submitted work. J.P.A. reports stock ownership from Jounce, Neon, BioAtla, Forty-Seven, Apricity, Polaris, Marker Therapeutics, Codiak, Kite Pharma, and consultant or advisor fees from Jounce, Neon, Amgen, Forty-Seven, Apricity, Polaris, Marker Therapeutics, Codiak, BioAlta LLC, Tvardi Therapeutics, and owns patents licensed to Jounce, Merck & BMS. M.T.T. reports personal fees from Myriad Genetics, Seattle Genetics and Novartis LLC, all outside the submitted work. C.N.S., A.R., and J.A.W. are co-inventors on US patent (PCT/US17/53.717) relating to the microbiome, outside of the current work. J.A.W. reports speaker fees from Imedex, Dava Oncology, Omniprex, Illumina, Gilead, MedImmune, and Bristol-Myers Squibb; consultant/advisor roles or advisory board membership for Roche-Genentech, Novartis, AstraZeneca, GlaxoSmithKline, Bristol-Myers Squibb, Merck/MSD, Biothera Pharma, and Microbiome DX; and receives clinical trial support from GlaxoSmithKline, Roche/Genentech, Bristol-Myers Squibb, and Novartis, all outside the current work. M.A.D. reports being the PI of research supported by grants to his institution from Myriad, AstraZeneca, Roche/Genentech, GlaxoSmithKline, Oncothyreon, and Sanofi-Aventis; consulting for NanoString; and advisory board participation for GlaxoSmithKline, Roche/Genentech, Novartis, Array, Bristol-Myers Squibb, Sanofi-Aventis, and Vaccinex. W.J.H. reports research support from Merck, Schering-Plough, GlaxoSmithKline, Bristol-Myers Squibb, and MedImmune, outside the current work. P.A.F. reports consulting for Gene+, outside of the current work. All other authors declare no competing interests.
