## [Peer Review File · Nature Communications]

Reviewers' comments:

Reviewer #1 (Remarks to the Author): Expertise in cancer immunology and T-cell heterogeneity

The manuscript by Futreal and colleagues aims at identifying spatial genomic and immunological heterogeneity in a melanoma metastasis progressive during anti-PD-1 treatment. By processing serial tumor sections for either IHC or genomic/proteomic analysis with further regional subsampling, which covered 67 regions of the tumor, they created an extensive map of immunogenomic profiling. Bulk analysis of a pre- and a post-treatment biopsy provided further information on temporal evolution of the tumor. While the manuscript addresses a crucial question – how extensive can spatial heterogeneity be throughout a tumor and how could this influence treatment response – and the effort to do this extensive analysis should be appreciated, there are some issues that need to be addressed before publication can be considered.

Major points:

1. The findings of this study are based on one tumor sample; however, conclusions are drawn for a more general patient population. This is difficult as the patient included in the study showed a highly unusual disease course with long-term survival despite never achieving an objective response to any treatment (indicating that there likely is some extent of immune control present in this patient, which notion is supported by the observed T cell clone persistent over time).
2. In addition, the extensive intratumoral heterogeneity observed in this patient may be caused by the multiple prior treatments she received. Therefore, many statements (e.g. that based on the clustering analysis multiple biopsies are needed to cover all tumor areas) may be true for this patient, but not necessarily applicable to other patients. To obtain more evidence, a similar analysis of a pretreatment tumor or on-treatment tumor after only one line of anti-PD-1 therapy would be important.
3. From a more technical aspect, it is unclear to me how well the heterogeneity in the 3rd dimension (along the axis in which the tumor has been sliced) has been taken into account. How comparable are the first and the last slide of the IHC slices (or if available of the slices used for genomics)? Based on the images provided, it seems that the slices and subregions vary quite a bit, e.g. the subregions in Fig 1C do not fit with sections 2, 4, 6, 8 in Fig 1B? Furthermore, could some tumor subregions be classified e.g. as 'core' based on the 2-dimensional assessment, but in fact become a 'margin' region in the 3rd dimension?
4. On a related note, it seems that many of the infiltrated samples do not have strong immune activation signatures in the RNA signatures (Fig 2B). Which cut-off has been used to classify an IHC sample as 'infiltrated'? Also, the samples that are identified to have similar immune composition using the Euclidean distance metrics in Fig E seem mostly to be samples with low RNA signature (e.g. the indicated samples 4A7 and 6A3, which actually differ in the IHC assessment).
5. How 'pure' are the subregions, e.g. if a sample is classified as 'margin', is the margin the dominant region in the sample or do for instance 2/3 of the sample contain 'core' tumor and the margin is just included in a corner of the subregion? Could this explain some of the heterogeneity measured? It would be helpful to include representative IHC images of the regions classified as 'core' and 'margin'. Furthermore, there are subregions in Fig. 2A classified as 'external', which are either differently labeled or missing in subsequent figures.
6. To gain further information on the persistent dominant T cell clone, the authors perform single cell sequencing. However, as this has been done in a distinct metastasis, it would be crucial to know whether this lesion was stable or whether it also progressed during aPD-1 treatment as this may lead to very distinct T cell phenotypes. In addition, as dysfunction is rather a gradual than a divergent process, the presence of this T cell clone in both activated and exhausted clusters may simply indicate distinct snapshots on a time axis than distinct priming events (e.g. activated vs previously activated cells as the authors mention later in the discussion). This should also be stated accordingly in the result section.

Minor points:

- Fig. 2B, Supp. Fig. 2A: what do the colors of the Immune score (ESTIMATE) indicate?

- P6 L144: is the reference to Fig 2B correct?
- Fig 2D: how do these protein modules correlate with other parameters (sample location, immune infiltrate?)
- Fig 3B: does the lower staining show CD8 (as indicated in the figure) or CD45 (as indicated in the figure legend)?
- Fig 4H is not mentioned in the text and indicated in the legend as Fig 4E.

Reviewer #2 (Remarks to the Author): Expertise in immunogenomics

Mitra et al. conduct a laudable comprehensive spatial immunogenomic profiling through careful gross dissection of an intractable melanoma metastasis on anti-PD-1 therapy. Although putative driver mutations are shared across the tumor's subregions, there is intratumoral heterogeneity of copy number alterations, gene expression signatures, and local immune infiltrates. One copy number variant in particular—gains of chromosome 7—was found in public bulk sequencing data to be associated with aPD1 non-responders; and chromosome 10 loss (including PTEN) was associated with subregions at the tumor periphery. Notably TCR repertoire profiling (including additional metastases and PBMCs) demonstrated substantial intra-tumoral heterogeneity of T-cell clones; as well as a T-cell clone that predominated in both the on-aPD1 and treatment-naïve metastases (spanning across 7 years).

Taken together, the authors' findings highlight the importance of multi-regional spatially-resolved analyses of tumor-immune interactions, and identify some putative mechanisms that may underlie aPD1-resistance in a subset of melanomas. The manuscript's methodology is sound and comprehensive. There are; however, several concerns that may encumber the manuscript.

1) The meticulous gross dissection of the lesion in order to permit such spatial resolution of analyses is commendable and as the foundation of the manuscript, could be better characterized for the readers as follows:

a) because the spatial coordinates/IDs of subregions (eg 2A10, 8A4) are a key component of the manuscript's analyses, it would be helpful if the spatial mapping from Fig 1B was greatly enlarged with improved resolution in order to provide the readers with better orientation.

b) additionally, because the spatial relationships are crucial to the manuscript, it would be helpful if the authors expanded their explanation of sample selection for their analyses:

i. e.g. was there a tumor viability cut-off for each methodology, in addition to the 80% tumor cell purity threshold for targeted sequencing in line 483 (for which the corresponding n of subregions that met that threshold should be included)?

ii. of the 67 subregions, why were only n=38 available for targeted-sequencing and methylation profiling, but n=46 were available for TCR sequencing and n=48 for RNAseq? A supplemental table/graphic might be useful to convey which sub-regions/tumors/PBMCs had which analyses. Presumably the analyzed conditions (margin vs. core, immune infiltrated vs. not) were well-represented in all genomic analyses?

iii. What were the n=6 subregions that underwent WES and how were they selected?

iv. As an example - The (very low-resolution) gross images of the fresh frozen sections and their corresponding H&Es in Fig 1B and hematoxylin counterstains from supp Fig 3 seem to suggest that samples 8A4, 8A5, 8A9, and 8A13 are overwhelmingly non-tumor soft tissue, and yet 8A4 was included in downstream analyses, whereas 8A5, 8A9, and 8A13 were not. Likewise in Fig 1E, 8A8 and 4A2 seem to lack the 6p, 20q, 6q, 9p, etc CNAs that characterize the melanoma - suggesting that these regions were also mostly non-tumor soft tissue. If non-tumor subregions are included in the analyses and grouped with tumor subregions, the corresponding results would likely be

skewed.

v. Additionally, multiple subregions from 8A are conspicuously missing from Fig 1E – could the authors please address?

2) The authors show enrichment of neutrophil activation signatures, particularly associated with chr7 gains, despite CIBERSORT analysis and ESTIMATE analysis of bulk sequencing data suggesting minimal infiltrating neutrophils. Additional data supporting and clarifying the putative roles of neutrophils would be helpful.

a. the highest concentration of neutrophils by CIBERSORT (supp table 3) and highest neutrophil immune signature z-scores by ESTIMATE (Fig 2B) seem associated with tumor section 8A, which was described as being highly necrotic. Can the authors comment further on whether the activation/accumulation of neutrophils is an infiltration specific to subregional molecular changes, or being detected as just a normal component of innate effector response to necrosis? Are the results the same if the analysis is done in between tumor sections, instead of chr7 gain vs chr7 stable subregions?

b. straightforward methods to confirm neutrophilic infiltration would include staining for neutrophils (eg stain for myeloperoxidase) of the FFPE sections and flow cytometric analysis of the dissociated lesion 1 and lesion 3 samples; and would help bolster the results.

c. did the authors apply the same CIBERSORT analysis to the bulk expression data from the TCGA cohort and 3x clinical cohorts to determine the neutrophil estimates for chr7 gain vs. chr7 stable regions? Those findings would also be of interest.

d. Because the lesion 1 and lesion 3 dissociated samples were sorted for either CD45+CD3+ or MCSP+ cell populations, neither of which should include neutrophils, presumably the same neutrophil activation signatures were not detected in their scRNAseq results?

3) The authors demonstrate the longitudinal TCR profiling of treatment-naïve, on-aPD1, and post-aPD1 lesions. In addition to sharing a key T-cell clonotype, the longitudinal coupling aspect of this manuscript would be substantially strengthened by evaluating whether the observed subregional copy number variants, SNVs, and methylation patterns were also shared with the pre-aPD1 lesion 1 and post-aPD1 lesion 3. For example, adding the data from lesions 1 and 3 to the CNA plot Fig1E (in addition to just chr10's data in sup Fig1), transcriptional hierarchical clustering in Fig2, gene set enrichment analysis in Fig3F, supFig2, sup Tables1&2, etc.

4) lines 190-209: the authors describe the putative ssGSEA and associated IHC findings qualitatively, but the results would be more compelling if displayed quantitatively with their corresponding statistics; e.g. fold-change for comparing WNT/B-catenin enrichment between the categorical phenotypes of margin vs. core and immune infiltrated vs. not; or goodness-of-fit testing in terms of the observed frequencies of pERK+ IHC and overlapping immune infiltrate.

5) The authors point out the limited-to-no objective response, presumably on radiography of the ventral wall lesion 2 leading up to resection there was no evidence of radiographic response to aPD1 that might suggest that the necrosis in section 8A was associated with pathologic treatment response to aPD1?

6) In reference to the claims in lines 385-387 and 393-395 regarding the lack of prior efforts to finely profile melanomas with spatial resolution, it might be helpful to re-frame the claims in the context of:

Thrane et al. Spatially Resolved Transcriptomics Enables Dissection of Genetic Heterogeneity in Stage III Cutaneous Malignant Melanoma. *Cancer Res.* 2018 Oct 15;78(20):5970-5979.
Which employs spatially-resolved scRNAseq to evaluate intratumoral heterogeneity across

metastases from multiple melanoma patients.

7) The limitations posed by spatially analyzing only a single small lesion from a single patient should be addressed.

8) Minor:

a. lines 238-241: it would be helpful if low-minimal and moderate-high immune cell content thresholds were defined.

b. Methods: the versions of all bioinformatics tools should be specified (eg MuTect, NetMHCpan, etc) for reproducibility and because outputs often vary slightly between versions.

c. line 488: best practices for removing duplicates, indel removal and recalibration should be referenced.

d. In addition to the 1x predicted neoantigen SNV shared across all tumor samples, it would be interesting to know how the remainder of the neoantigen repertoire is distributed across tumor subregions and how they are associated with the distribution of T-cell clonotypes; with the noted limitation that MHC binding affinity algorithms are severely handicapped.

Reviewer #3 (Remarks to the Author): Expertise in genomics

The manuscript by Mitra et al. reports a very deep characterization of a single patient by performing immunogenomic analyses on 67 intra-tumor sub-regions of a PD-1 inhibitor resistant tumor, and 2 additional melanoma metastases. The work is very well presented and would be have great interest to the readership. The paper has some tantalizing findings, especially those related to chr 7 gain which were explored in a larger dataset.

Comments/questions to the authors:

1. Many of the conclusions from the paper are from a single individual, who was a long-term survivor who was heavily treated. The n of 1 is a potential weakness. Inclusion in the discussion of whether this sample is really representative of other cases needs to be included.

2. Was the degree of genomic heterogeneity (SNV and CNA) across the heavily pre-treated tumor surprising? Would you expect more clonal section in such a pre treated sample?

3. Does the RNAseq indicate which genes are being affected by the copy number changes at chromosome 10 (tumor margin events)?

4. The ch7 finding is intriguing. Do the low counts of immune cells (from RNAseq) within chr7 regions correlate with IHC assessment of these regions? Even though immune cell counts were low, were the proportion of immune cells similar to other sites? What genes were in the 'positive neutrophil activation' and 'negative neutrophil activation' signatures?

The inclusion of the TCGA supports a role for chr7. Can the TCGA RNAseq also under deconvolution to see if there was a similar count or proportion of immune cells in the TCGA data?

5. The authors show that the neutrophil signature is associated with response to immunotherapy in public datasets. Was chr7 gain also associated with response to checkpoint blockade?

6. There is a large amount of omic work (RNAseq, DNaseq, methylation, TCR) from many small samples that may have been technically challenging. Can the authors summarise the data quality, ie:

a. The nucleotide yield must have been sufficient for these analyses, what was the yield of DNA and RNA for each site and the quality of the nucleotides (eg RIN score)?

b. What read depth was the RNA sequenced?

c. The TCR sequencing results seem highly variable across the tumor. Could this be due to under sampling the TCR at each site?

7. To support the conclusion that the common epitope is driving a persistent T-cell response, is it

possible to validate the immunogenicity of the ZDNNC17 p.H507Y neoantigen in vitro?

Minor comments

- It would be helpful to include more information on the figures or in the legends in terms of axis labels and legends so the values are clear, for example Fig 5A and B what is the value and how were the scores generated, also what is the value of 'size' in Fig 5C etc. Fig 1E include a scale indicating level of CNA.
- Ensure all methods described were used in the paper. For example the single cell methods include assessment of CNA using C

NCOMMS-19-19214

Reviewer #1 (Remarks to the Author): Expertise in cancer immunology and T-cell heterogeneity

The manuscript by Futreal and colleagues aims at identifying spatial genomic and immunological heterogeneity in a melanoma metastasis progressive during anti-PD-1 treatment. By processing serial tumor sections for either IHC or genomic/proteomic analysis with further regional subsampling, which covered 67 regions of the tumor, they created an extensive map of immunogenomic profiling. Bulk analysis of a pre- and a post-treatment biopsy provided further information on temporal evolution of the tumor. While the manuscript addresses a crucial question – how extensive can spatial heterogeneity be throughout a tumor and how could this influence treatment response – and the effort to do this extensive analysis should be appreciated, there are some issues that need to be addressed before publication can be considered.

Major points:

1. The findings of this study are based on one tumor sample; however, conclusions are drawn for a more general patient population. This is difficult as the patient included in the study showed a highly unusual disease course with long-term survival despite never achieving an objective response to any treatment (indicating that there likely is some extent of immune control present in this patient, which notion is supported by the observed T cell clone persistent over time).

We thank the reviewer for this observation and indeed agree that the findings in this study are primarily motivated by deep profiling of one lesion, together with comparative analyses as feasible in pre- and post-immunotherapy treatment longitudinal metastases. To the extent that we aimed to highlight the sheer extent of local microenvironmental heterogeneity even within single tumors, even though it is based on one tumor, our spatial profiling reveals the sensitivity of immunogenomic characteristics to regional features across the numerous sampled tumor volumes. Acknowledging differences in patient cohorts, we sought to explore whether our conclusions were more generally applicable considering the now frequent use of targeted therapy and immunotherapy in metastatic melanomas of all clinical presentations. Through these analyses, we indeed found that certain features observed initially in our sub-regional analysis of one tumor had correlates in a broader patient population, suggesting that there is robustness to the underlying biology that we describe.

Regarding this patient's clinical history, we too were initially interested in large part because of her history of multiple systemic therapies and long-term survival, and whether deep genomic profiling could shed light on this phenomenon. However we do not feel that such a presentation is as uncommon as generally thought. The survival curves for stage IV melanoma patients published with the AJCC 7th edition (Balch et al, JCO 2009;27(36):6199-6206) clearly indicate that a substantial proportion of patients with non-visceral metastatic disease (skin, subcutaneous or distant nodes; stage IVA) or to a lesser extent patients with at worst pulmonary metastatic disease (stage IVB) survive long-term; approximately 20% and 10%, respectively, being alive at 10 years following first diagnosis of metastatic disease. Importantly, this represents a patient population that pre-dates modern survival-prolonging therapies. Considering this baseline prevalence, the current availability of disease-modifying therapies, and the effect of survivorship bias to enrich this patient population in clinic attendees, we do not feel that such long-term surviving patients with low to moderate burden disease should be considered rare or highly unusual in the current reality of standard clinical practice.

2. In addition, the extensive intratumoral heterogeneity observed in this patient may be caused by the multiple prior treatments she received. Therefore, many statements (e.g. that based on the clustering analysis multiple biopsies are needed to cover all tumor areas) may be true for this patient, but not necessarily applicable to other patients. To obtain more evidence, a similar analysis of a pretreatment tumor or on-treatment tumor after only one line of anti-PD-1 therapy would be important.

This is an excellent point, and we agree that the extensive ITH observed in this particular tumor could be a result of prior treatment exposures, including targeted agents and prior immunotherapy. Two moderating considerations arise: firstly, all prior lines of therapy failed to produce objective responses in the lesion sampled, thus it is unclear whether one should expect that those lines of prior therapy had meaningfully sculpted the intratumoral heterogeneity observed in this lesion; secondly, if prior therapies did influence or drive the observed heterogeneity in this tumor, it could be argued that this would generally lead to a reduction in the absolute level of heterogeneity (rather than an increase) as a result of clonal pruning and selective pressures.

In the context of anti-cancer immunity, a great deal of dynamic interaction between tumor cells, stroma and immune cells has already occurred spontaneously even at the time the patient is first diagnosed, and this undoubtedly contributes to the immune permissive or immune resistant status of the tumor prior to any treatment being initiated. Thus, a priori, it is not possible to know whether any given tumor may show high or low evidence of spatial heterogeneity at an immune or genomic level. We also know that reliable biomarkers of response and resistance to immune checkpoint blockade are still lacking, and the poor performance of anything other than outlier-high levels of tumor mutational burden or PD-L1 expression (both useful if high, but non-discriminatory for response at moderate/low levels) may reflect an artifact of temporal or spatial heterogeneity. Our intent from presenting the clustering analysis and suggesting that multiple tumor regions should be sampled in order to comprehensively represent the immune status of the tumor was to reflect this, however this may very well not be necessary for all melanoma tumors. We agree that a similar analysis of pre-treatment samples and additional drug-exposed samples would be of value to address this question and should be explored in sizeable cohorts in the future, however this represents a substantial undertaking that is beyond the scope of the current manuscript. We have added the following text to the discussion:

“Whilst the degree of immunogenomic spatial heterogeneity in any given tumor mass cannot yet be predicted non-invasively, the spontaneous nature of immune-tumor interactions implies that relevant spatial heterogeneity will be found, irrespective of prior therapeutic exposures.”

3. From a more technical aspect, it is unclear to me how well the heterogeneity in the 3rd dimension (along the axis in which the tumor has been sliced) has been taken into account. How comparable are the first and the last slide of the IHC slices (or if available of the slices used for genomics)? Based on the images provided, it seems that the slices and subregions vary quite a bit, e.g. the subregions in Fig 1C do not fit with sections 2, 4, 6, 8 in Fig 1B? Furthermore, could some tumor subregions be classified e.g. as ‘core’ based on the 2-dimensional assessment, but in fact become a ‘margin’ region in the 3rd dimension?

We apologize for the confusion associated with exactly how the spatial analysis was performed as it is a difficult process to describe! We have modified the methods text describing this, and adjusted Figure 1B and Supplementary Figure 1 to make this clearer.

Essentially, the gross tumor was serially sectioned perpendicularly to its largest axis resulting in 8 slices of 2mm to 3 mm thickness each. Lateral (left/right) orientation was preserved by differential painting of the outside surface prior to sectioning with red or blue ink. Odd-numbered

slices were used for IHC after formalin fixation and embedding in paraffin, whilst even-numbered slices were preserved as frozen sections embedded in OCT and used for genomic and protein (RPPA) characterization.

The first and last slices used for IHC would hence vary substantially based on their separation by the entire tumor mass present in between. A major source of confusion may have arisen from the fact that the sub-regional numbering is not shared between the slices. This was due to the irregular shape of the mass such that successive slices presented markedly different cross-sections of tumor that were more appropriately subdivided on a section-by-section basis. Each slice (frozen plus adjacent FFPE) therefore has a unique numbering pattern although in general numbering proceeded bottom-to-top and right-to-left. We have updated the methods section to clarify these procedures, most particularly:

“Due to the variation in tumor shape throughout three-dimensional space, each tumor slice presented a distinct cross-sectional area and thus a unique grid was applied to each frozen tumor slice (slice n) and the immediately adjacent FFPE slices (slices $n-1$ face B and $n+1$ face A). Thus, whilst sub-region numbering generally proceeded bottom-to-top and right-to-left, specific slice sub-region numbering is not directly comparable between slices.”

When assigning sub-region classes to frozen sections, we accounted for three-dimensional variation throughout the slice by considering the FFPE slices both above and below (when available) and arrived at a consensus call to categorize tumors as core or margin. Hence, we did not consider core/margin within each slice by itself, but rather as an aggregated tumor mass in 3-dimensions, and therefore it is possible that a region presenting as “core” in a single two-dimensional section may have been categorized as “margin” due to the presence of a tumor margin component in the associated tumor volume. The methods text has been extended as follows:

“5.1) Designation of sub-regions as located at the tumor core or margin was performed by inspection of all regions annotated on SOX10-stained IHC slides to infer a volumetric estimate of tumor content and location. Three-dimensional variation throughout the frozen slices was accounted for by considering the immediately adjacent FFPE slices (ie: above AND below, when both were available) in order to arrive at a consensus call to categorize tumors as core or margin.”

4. On a related note, it seems that many of the infiltrated samples do not have strong immune activation signatures in the RNA signatures (Fig 2B). Which cut-off has been used to classify an IHC sample as ‘infiltrated’? Also, the samples that are identified to have similar immune composition using the Euclidean distance metrics in Fig E seem mostly to be samples with low RNA signature (e.g. the indicated samples 4A7 and 6A3, which actually differ in the IHC assessment).

The reviewer raises a very pertinent point about IHC assessment because in the context of immune infiltration of tumor masses, even digitally-quantified IHC requires a degree of manual curation to account for distribution of marker staining (eg: CD45) and for this specific study, consensus estimates using adjacent FFPE sections to interpolate immune content within the frozen sections used for genomic analyses. Thus it is at best semi-quantitative which is why we chose to represent this primarily as a categorical variable. The following additional methods text has been added:

5.2) Immune infiltrates were evaluated using CD45-LCA positive cell density measured by digital image quantification using the Aperio ImageScope software. CD45+ density was then categorized as low, medium or high by binning into

lower, middle, or upper tertiles considering all regions analyzed. Several additional factors required consideration before arriving at a final semi-quantitative categorization as having focal, low, moderate or high immune infiltration: between-slide variation in staining efficiency, spatial distribution of immune cell content (focal, broad, intra/peri-/extra-tumoral), and a consensus estimate for genomic frozen slices by considering the adjacent FFPE sections on both sides for which the higher immune content was assigned precedence.

In Figure 2E we attempted to identify samples with similar immune composition which is conceptually distinct from samples having high or low immune cell densities by a more crude measure such as CD45-positive staining on IHC. Therefore, we expected to see differences between the findings presented in Figure 2E and what might be predicted based solely on reductive IHC or RNA signatures such as the ESTIMATE immune score. For example, regions 6A10, 6A11 and 6A15 may have generally lower overall immune content, however the composition of this content and thus sample-level clustering pattern, shares similarities such as a predominance of NK and CD8 T cell signatures and relative lack of Type-1 IFN and myeloid signatures, as seen in Figure 2B. As outlined above, the RNA assessment and IHC assessment may also differ due to their derivation from adjacent but nonetheless non-identical slices of the tumor which necessarily is subject to compositional variation due to the irregular shape of the tumor mass; this may have contributed to the apparent disconnect seen in regions 4A7 and 6A3. We have added a clarification to the following sentence:

“Using sample-wide Euclidean distance metrics to connect samples with highly similar immune composition based on immune deconvolution rather than reductive immune scores or overall immune cell densities, we found that similar immunophenotypes...”

5. How ‘pure’ are the subregions, e.g. if a sample is classified as ‘margin’, is the margin the dominant region in the sample or do for instance 2/3 of the sample contain ‘core’ tumor and the margin is just included in a corner of the subregion? Could this explain some of the heterogeneity measured? It would be helpful to include representative IHC images of the regions classified as ‘core’ and ‘margin’. Furthermore, there are subregions in Fig. 2A classified as ‘external’, which are either differently labeled or missing in subsequent figures.

Core regions were defined as those containing no margin element on any side. Analytically, the majority of genomic regions either clearly did, or were inferred to contain some margin element based on the shape of the tumor above and below the genomic slice, which raised the exact question the reviewer has raised. To address the potential confounding effect of variable net tumor cell content when comparing core regions with margin regions, we evaluated these differences using two strategies; firstly by comparing core regions with all margin regions, and secondly by comparing core regions with only those margin regions estimated to have at least 80% tumor cell content based on SOX10 immunostaining, and found these results to be essentially concordant. We also performed immune high versus low analysis within margin regions both including either all margin samples or only margin samples containing at least 30% tumor cells, again finding consistent results overall.

We have included a supplemental figure to illustrate representative core and margin regions, and also thank the reviewer for identifying the inconsistency in the sample location annotation track shown on the original Fig. 2A, which has been corrected.

6. To gain further information on the persistent dominant T cell clone, the authors perform single cell sequencing. However, as this has been done in a distinct metastasis, it would be crucial to know whether this lesion was stable or whether it also progressed during aPD-1 treatment as this may lead to very distinct T cell phenotypes. In addition, as dysfunction is rather a gradual

than a divergent process, the presence of this T cell clone in both activated and exhausted clusters may simply indicate distinct snapshots on a time axis than distinct priming events (e.g. activated vs previously activated cells as the authors mention later in the discussion). This should also be stated accordingly in the result section.

The gluteal metastasis which was used for single cell sequencing was performed on a mass that was slowly progressive towards the end, and after, anti-PD-1 blockade. To clarify this we have added the following text to the legend for Figure 1A:

“Molecularly-profiled lesions are indicated: index left lower lobe (LLL) lung metastasis (lesion 1), progressing ventral abdominal wall mass (lesion 2) and slowly progressing right gluteal mass (lesion 3).”

We agree with the reviewer that the acquisition of a dysfunctional T cell phenotype is effectively a gradual, albeit somewhat stochastic, process, however in this case we have leveraged TCR sequencing data to describe a T cell clonotype (at the amino acid level) that actually represents a population of unquestionably distinct T cell clones (at the DNA level) arising from distinct TCR gene rearrangement events. Furthermore, these T cell clones are clearly evident across a timespan of nearly a decade. Thus whilst we agree that a single activated T cell population may evolve into a mixed population of activated and exhausted/dysfunctional cells following a single priming event, this would not be compatible with the particular combination of independent TCR rearrangement clones and time span that we describe, hence we postulate that it is likely to indicate repeated priming events. We have adjusted the wording in the results to clarify this point:

“The detection of multiple clones at nucleotide level expressing a synonymous CDR3 amino acid sequence, their persistence over nearly a decade, and simultaneous presence of both activated and exhausted phenotypes suggests that this T cell population arose from multiple independent T-cell priming events rather than functional divergence following a single more recent priming/activation event.”

Minor points:

- Fig. 2B, Supp. Fig. 2A: what do the colors of the Immune score (ESTIMATE) indicate?

We apologize for not clearly indicating that the color track utilizes an analogous violet-through-orange gradient as in the heatmap. The labelling has been adjusted accordingly.

- P6 L144: is the reference to Fig 2B correct?

Indeed no, the reference should have been to Fig. S2A which is now Supp. Fig. 3A. We have corrected this error.

- Fig 2D: how do these protein modules correlate with other parameters (sample location, immune infiltrate?)

The protein set represented on the RPPA showed no statistically significant differential expression according to the sub-region location at either core or margin sites. When considering immune infiltrate, we observed several differentially-expressed proteins (at $FDR < 0.10$) with enrichments that paralleled previously-observed co-expression associations. For example, MYH11, LCK and PTK2 were all highly co-expressed and were additionally enriched in immune-high sub-regions. Conversely, SOX2 and GLS were similarly highly co-expressed and enriched in immune-low sub-regions, consistent with the notion that immune-low sub-regions generally have higher tumor cell content due to less cellular ‘dilution’. To indicate these additional results, we have added annotation to Figure 2D indicating which proteins shown in the co-expression matrix are statistically enriched in immune-high (starred and red) or immune-low (starred and blue) sub-regions.

- Fig 3B: does the lower staining show CD8 (as indicated in the figure) or CD45 (as indicated in the figure legend)?

The lower panel in Fig. 3B is CD8 staining; the figure legend has been corrected accordingly.

- Fig 4H is not mentioned in the text and indicated in the legend as Fig 4E.

Thank you for identifying this mistake. The text citation on page 12 incorrectly referred to Fig. 4F-G and this has been corrected to Fig. 4G-H. The legend has been corrected to indicate panel (H) rather than (E) twice.

Reviewer #2 (Remarks to the Author): Expertise in immunogenomics

Mitra et al. conduct a laudable comprehensive spatial immunogenomic profiling through careful gross dissection of an intractable melanoma metastasis on anti-PD-1 therapy. Although putative driver mutations are shared across the tumor's subregions, there is intratumoral heterogeneity of copy number alterations, gene expression signatures, and local immune infiltrates. One copy number variant in particular—gains of chromosome 7—was found in public bulk sequencing data to be associated with aPD1 non-responders; and chromosome 10 loss (including PTEN) was associated with subregions at the tumor periphery. Notably TCR repertoire profiling (including additional metastases and PBMCs) demonstrated substantial intratumoral heterogeneity of T-cell clones; as well as a T-cell clone that predominated in both the on-aPD1 and treatment-naïve metastases (spanning across 7 years).

Taken together, the authors' findings highlight the importance of multi-regional spatially-resolved analyses of tumor-immune interactions, and identify some putative mechanisms that may underlie aPD1-resistance in a subset of melanomas. The manuscript's methodology is sound and comprehensive. There are; however, several concerns that may encumber the manuscript.

1) The meticulous gross dissection of the lesion in order to permit such spatial resolution of analyses is commendable and as the foundation of the manuscript, could be better characterized for the readers as follows:

a) because the spatial coordinates/IDs of subregions (eg 2A10, 8A4) are a key component of the manuscript's analyses, it would be helpful if the spatial mapping from Fig 1B was greatly enlarged with improved resolution in order to provide the readers with better orientation.

We agree with the reviewer that the presentation of this central component of our study could be better represented and have re-designed Fig. 1B to include higher-quality images of the sectioning process and sub-region division, and Supplementary Fig. 1 (previously Supp. Fig. 3) now includes indicative regions mapped onto each IHC slice with the SOX10 stain. We have also revised the associated methodology text to emphasize that due to the variation in tumor shape throughout three-dimensional space and therefore between tumor slices, the sub-region divisions are not intended to be identical for each frozen slice:

“Due to the variation in tumor shape throughout three-dimensional space, each tumor slice presented a distinct cross-sectional area and thus a unique grid was applied to each frozen tumor slice (slice n) and the immediately adjacent FFPE slices (slices n-1 face B and n+1 face A). Thus, whilst sub-region numbering generally proceeded bottom-to-top and right-to-left, specific slice sub-region numbering is not directly comparable between slices.”

b) additionally, because the spatial relationships are crucial to the manuscript, it would be helpful if the authors expanded their explanation of sample selection for their analyses:

i. e.g. was there a tumor viability cut-off for each methodology, in addition to the 80% tumor cell purity threshold for targeted sequencing in line 483 (for which the corresponding n of subregions that met that threshold should be included)?

The tumor viability for all genomic analyses including targeted sequencing and methylation as well as RNA sequencing was estimated from corresponding IHC samples and was set at a threshold of 80% viable tumor purity. Correspondingly, samples with DNA integrity numbers (DIN) greater than 7 were used for targeted panel sequencing and EPIC array methylation profiling. RNAseq was performed on samples with a minimum RNA integrity number (RIN) of 5.5 except for two cases (6A10 and 8A3) with RINs greater than 3. Additionally, a minimum of 700ng of RNA were required for all samples undergoing RNAseq. We have added these details to the methods section “Nucleic acid extraction”.

ii. of the 67 subregions, why were only n=38 available for targeted-sequencing and methylation profiling, but n=46 were available for TCR sequencing and n=48 for RNAseq? A supplemental table/graphic might be useful to convey which sub-regions/tumors/PBMCs had which analyses. Presumably the analyzed conditions (margin vs. core, immune infiltrated vs. not) were well-represented in all genomic analyses?

The n of 67 represents all sub-tumoral samples from the tumor mass. However, different sub-regions were utilized for molecular and immunohistological profiling. For RNAseq, n=48 passed QC thresholds (described above) and n=47 were evaluable after annotating sample site/tumor content. For TCR sequencing n=46 samples produced sufficiently high numbers of templates to permit further analysis. Targeted sequencing and methylation profiling were determined based off pathological purity estimates of greater than 20% viable tumor. We appreciate the reviewer’s comment regarding a supplemental table to highlight what profiling each sample underwent and have now added this as Supplementary Table 1. For differential expression analysis, we obtained 10 core samples and 37 samples from the margin. Additionally, in terms of immune infiltration 16 samples had low to focal immune infiltration while 31 samples had moderate to high immune infiltration.

iii. What were the n=6 subregions that underwent WES and how were they selected?

Sub-regions for WES were selected based on the dominant TCR clones that were detected in the regions. Three regions with the most dominant TCR clone, three regions with the second most prevalent clone and finally two regions with lower tumor purity were selected. From these eight sub-regions that were selected to undergo WES (2A2, 2A10, 4A1, 4A2, 4A11, 6A3, 6A16, 8A8) as well as the treatment naïve lung metastasis (lesion 1) and post-PD-1 blockade gluteal metastasis (lesion 3), we performed library prep and QC and only selected samples that had a DIN greater than 7.5 (ultimately pre-treatment lesion, regions 2A2, 4A11, 6A3, 6A16, and post-PD-1 blockade lesion).

iv. As an example - The (very low-resolution) gross images of the fresh frozen sections and their corresponding H&Es in Fig 1B and hematoxylin counterstains from supp Fig 3 seem to suggest that samples 8A4, 8A5, 8A9, and 8A13 are overwhelmingly non-tumor soft tissue, and yet 8A4 was included in downstream analyses, whereas 8A5, 8A9, and 8A13 were not. Likewise in Fig1E, 8A8 and 4A2 seem to lack the 6p, 20q, 6q, 9p, etc CNAs that characterize the melanoma – suggesting that these regions were also mostly non-tumor soft tissue. If non-tumor subregions are included in the analyses and grouped with tumor subregions, the corresponding results would likely be skewed.

We apologize for the low resolution of the gross images and have upgraded these in the new version of Figure 1B but we are limited by the quality of images taken at that time. As evident in the new Supplementary Fig. 1, large areas of slice 8 contain low amounts of (or in some regions no) viable tumor, however we sought to characterize the entirety of the tumor mass, including contained and adjacent stromal and immune compartments that are necessarily present to varying extents in all translationally-studied samples. We feel that even subjacent stromal content forms an integral part of the dysfunctional tumor microenvironment and thus chose not

to aggressively exclude them from evaluation. However, we do agree that tumor content is a particular consideration for the interpretation of transcriptomic data (when evaluating tumor cell-derived signatures) thus when exploring core vs margin differences, we performed an analysis restricted to samples of comparably high estimated tumor content and found similar results (see response to Reviewer 1, Q5).

In the process of sample preparation and downstream analysis, certain samples (particularly from slice 8) had to be removed due to low yield of RNA/DNA consistent with the presence of necrotic tissue. Sample 8A4 contained viable tumor DNA and hence was selected for sequencing whilst other regions did not. We agree that certain sub-regions (e.g.: 8A8 and 4A2) lacked the CNAs characterizing this melanoma, consistent with low viable tumor content in these samples and tumor margin location. However, with deep targeted sequencing (1000x) we were nonetheless able to identify the characteristic $NRAS^{Q61R}$ mutation and the acquired $BRAF^{G421R}$ mutations seen in this particular tumor mass, indicative of the presence of a measurable amount of tumor cell content.

v. Additionally, multiple subregions from 8A are conspicuously missing from Fig 1E – could the authors please address?

Multiple subregions from slice 8 were not included for downstream analyses due to necrotic tissue and low viable DNA yield. Initial sample selection was performed using the matching IHC slide to determine samples with greater than 80% estimated tumor content. Following this, samples were selected which passed QC thresholds and DIN greater than 7 or a RIN greater than 5.5.

Additionally, we must reiterate that due to the tumor mass being irregularly-shaped in three dimensions, rather than a perfectly regular rectangular prism, it was not possible to sub-divide each frozen slice according to the exact same grid. Thus, the regions that appear to be conspicuously missing from slice 8 relative to other slices are not missing; each (frozen) slice is simply uniquely sub-divided. We have updated the methods text and incorporated sub-region gridding to Supplementary Figure 1 to assist the reader in their understanding of this.

2) The authors show enrichment of neutrophil activation signatures, particularly associated with chr7 gains, despite CIBERSORT analysis and ESTIMATE analysis of bulk sequencing data suggesting minimal infiltrating neutrophils. Additional data supporting and clarifying the putative roles of neutrophils would be helpful.

a. the highest concentration of neutrophils by CIBERSORT (supp table 3) and highest neutrophil immune signature z-scores by ESTIMATE (Fig 2B) seem associated with tumor section 8A, which was described as being highly necrotic. Can the authors comment further on whether the activation/accumulation of neutrophils is an infiltration specific to subregional molecular changes, or being detected as just a normal component of innate effector response to necrosis? Are the results the same if the analysis is done in between tumor sections, instead of chr7 gain vs chr7 stable subregions?

We agree with the reviewer that the regional presence of neutrophils raises the question of cause versus effect in this context. In this study, the neutrophil signatures, as calculated from MCP-counter and CIBERSORT, were found to be enriched in the necrosis-rich slice 8. These signatures were based on the expression of gene sets that were used to classify neutrophils in the LM22 dataset from CIBERSORT and the neutrophil markers in MCP-counter (listed in Supplementary Table 7). We have also performed immunostaining for neutrophils as per the following question, which is consistent with the observation of high neutrophil densities in necrotic regions (representative images in Supplementary Figure 3D). From the cross-sectional

(i.e.: not longitudinal) data available to us it is unfortunately not possible to know whether neutrophils are present because these tumor areas were necrotic, or whether these tumor areas were necrotic at least in part because of neutrophil activity.

The neutrophil activation/degranulation signature is distinct from the neutrophil density signatures enumerated by MCP-counter and CIBERSORT, being calculated using genes that define neutrophil activation as based on gene ontology (GO) term analysis (provided in Supplementary Table 8). These signatures were found to be enriched in the chromosome 7 gain sub-regions.

To address between-tumor slice variation, we performed differential expression between different slices (slice 2 vs all other slices, slice 4 vs all other slices, slice 6 vs all other slices and slice 8 vs all other slices), which did not identify significant pathway enrichments when querying either the GO or KEGG pathway databases.

b. straightforward methods to confirm neutrophilic infiltration would include staining for neutrophils (eg stain for myeloperoxidase) of the FFPE sections and flow cytometric analysis of the dissociated lesion 1 and lesion 3 samples; and would help bolster the results.

We only have FFPE material remaining from lesion 2, and have performed CD15 staining as recommended by our pathologist. As expected, there was low-level scattered neutrophil presence throughout most regions, and particularly high-density neutrophil content in the necrotic regions of slice 7 (eg: 7A9, 7A15, 7A16, 7B10, 7B11) together with a locally high background staining due to CD15+ subcellular debris. It is not possible to determine whether neutrophil accumulation in necrotic areas is cause or effect considering the static nature of this IHC. We have incorporated CD15 staining results into the IHC correlations plot shown in Fig. 2C, and have added the following text to the results:

“A notable exception was particularly high T and B cell signatures in multiple samples of section 8 (8A6, 8A7, 8A8, and 8A13), which was highly necrotic and displayed heavy neutrophil infiltration on matched FFPE slices (CD15 stain; Supplementary Fig. 3D) although this could not be determined as the cause or consequence of necrosis.”

Unfortunately a single cell suspension was never prepared from lesion 1, hence we could not profile it further. Additionally there was only limited cell suspension of lesion 3 which has been exhausted.

Regarding the specific question of how neutrophil densities relate to chr7 gain regions, outside of necrotic tumor areas in which CD15 staining was heavy and potentially complicated by background staining, neutrophil densities were confirmed to be variable but generally low throughout the remainder of the tumor including chr7 gain regions, which is why we interrogated neutrophil function genesets to infer putative neutrophil activity. We have added the following clarification to the text:

“However, whilst overall neutrophil counts derived from transcriptome data were low and this was consistent with generally low neutrophil densities identified by CD15 immunostaining of corresponding FFPE sections, interrogation of neutrophil activation gene sets...”

c. did the authors apply the same CIBERSORT analysis to the bulk expression data from the TCGA cohort and 3x clinical cohorts to determine the neutrophil estimates for chr7 gain vs. chr7 stable regions? Those findings would also be of interest.

CIBERSORT analysis was not performed previously on the bulk expression data from TCGA or the clinical cohorts. We have now performed CIBERSORT analysis on all datasets and found that neutrophils (as determined from the LM22 signature) were not found to be more enriched in the chromosome 7 gain samples or in melanoma patients that failed to respond to ICB. From the CIBERSORT analysis, the Van Allen and Riaz datasets have higher neutrophil proportions, however this did not reach statistical significance. The Hugo dataset displayed an opposite trend with non-responders having fewer neutrophil proportions. These results have been added as Supplementary Figure 8. It remains possible that regional variation in neutrophil density exists at a sub-tumoral level, however considering the bulk analyses of all tumors across these cohorts it may not be possible to identify biologically-relevant differences in this manner.

d. Because the lesion 1 and lesion 3 dissociated samples were sorted for either CD45+CD3+ or MCSP+ cell populations, neither of which should include neutrophils, presumably the same neutrophil activation signatures were not detected in their scRNAseq results?

The post-PD-1 lesion (lesion 3) was sorted for CD45+CD3+ for T-cells and MCSP+ for malignant cells. Additionally, the tumor digest was stored frozen prior to analysis, and it is well-known that granulocytes effectively do not survive freeze-thaw cycles. We did not obtain an enrichment for neutrophil activation in the sc-RNAseq results, and this was expected. The treatment-naïve lesion (lesion 1) did not have available single-cell suspension.

3) The authors demonstrate the longitudinal TCR profiling of treatment-naïve, on-aPD1, and post-aPD1 lesions. In addition to sharing a key T-cell clonotype, the longitudinal coupling aspect of this manuscript would be substantially strengthened by evaluating whether the observed subregional copy number variants, SNVs, and methylation patterns were also shared with the pre-aPD1 lesion 1 and post-aPD1 lesion 3. For example, adding the data from lesions 1 and 3 to the CNA plot Fig1E (in addition to just chr10's data in sup Fig1), transcriptional hierarchical clustering in Fig2, gene set enrichment analysis in Fig3F, supFig2, sup Tables1&2, etc.

We performed whole-exome sequencing and TCR-sequencing on the pre-treatment and post-PD-1 treated lesion. We leveraged the SNVs and mutation data to track mutations across multiple metastases over time. Additionally, we predicted putative neoantigens using mutations that were shared across all the samples. Through this process, we narrowed down one neoantigen that was found across all the timepoints/samples and had a low IC50. We have updated the CNA plot to include the data from lesions 1 and 3 as suggested by the reviewer and incorporated this into Supplementary Figure 2. We do not have RNA-seq or methylation data from lesions 1 or 3, and thus cannot include these in transcriptional clustering or gene set enrichment analysis. We agree with the reviewer that complete sets of all data types from all lesions would be an impressive resource, and it would be valuable to target such comprehensive profiling in future studies, funding and samples permitting.

4) lines 190-209: the authors describe the putative ssGSEA and associated IHC findings qualitatively, but the results would be more compelling if displayed quantitatively with their corresponding statistics; e.g. fold-change for comparing WNT/B-catenin enrichment between the categorical phenotypes of margin vs. core and immune infiltrated vs. not; or goodness-of-fit testing in terms of the observed frequencies of pERK+ IHC and overlapping immune infiltrate.

We agree that enhanced statistical rigor is desirable in all analyses where possible. In this case, we performed ssGSEA specifically to reveal the underlying regional heterogeneity that could otherwise be lost in grouped analyses, thus adding fold-change and statistical data for these Hallmark gene sets by grouped parameters like region site (core vs margin) or immune infiltrate (low vs high) essentially negates the value of single-sample GSEA. Nonetheless, we have performed a tailored version of differential expression analysis by applying linear models to this data to calculate fold-changes and statistics using empirical Bayes' methods and provide these

as Reviewer-only Tables R1 and R2. As expected, gene sets generally displayed low fold changes (all <2-fold for core vs. margin and several only marginally above 2-fold for immune low vs. high) when analyzed in this way and thus did not lead to meaningful biological insight. Additionally, where appropriate we have explicitly added statistical data to several key results in this section.

Multiplex IHC would be the ideal methodology to more quantitatively assess the co-localization of tumor cell pERK positivity and infiltrating/adjacent immune cells (e.g.: CD45LCA+). We performed singlet IHC which for logistical reasons involved cutting slides on separate occasions, including re-facing of several blocks between the times of CD45 and pERK staining. Due to the change in shape of the tumor throughout the depth of each block, it is unfortunately not feasible to perform co-localization studies of these stains using the available data.

5) The authors point out the limited-to-no objective response, presumably on radiography of the ventral wall lesion 2 leading up to resection there was no evidence of radiographic response to aPD1 that might suggest that the necrosis in section 8A was associated with pathologic treatment response to aPD1?

This is certainly possible. It is clear that radiographic responses do not entirely equate to pathologic responses, and indeed quantitative imaging analysis criteria have now been modified for immune-oncology therapies to account for the concept of pseudo-progression in which simple enlargement of a tumor mass may reflect a non-malignant process such as tumoral inflammation or regional necrosis. Nonetheless, pseudo-progression affects a small minority of patients/lesions and requires tumor shrinkage on longitudinal imaging before arriving at the retrospective diagnosis of pseudo-progression, which was not observed in this patient. The necrosis seen in this lesion on a pathologic level was relatively superficial which, combined with the relatively small overall tumor mass, would not be expected to be evident on standard imaging.

6) In reference to the claims in lines 385-387 and 393-395 regarding the lack of prior efforts to finely profile melanomas with spatial resolution, it might be helpful to re-frame the claims in the context of:

Thrane et al. Spatially Resolved Transcriptomics Enables Dissection of Genetic Heterogeneity in Stage III Cutaneous Malignant Melanoma. *Cancer Res.* 2018 Oct 15;78(20):5970-5979.

Which employs spatially-resolved scRNAseq to evaluate intratumoral heterogeneity across metastases from multiple melanoma patients.

We thank the reviewer for suggesting this very pertinent addition to our discussion, which we have included as follows:

“Thrane and colleagues performed a proof-of-principle high-resolution spatial transcriptomics analysis of four lymph node metastases obtained from patients with stage III melanoma, finding evidence of variably distinct gene expression profiles between regions of tumor, lymphoid tissue, and an apparent transition zone that may have represented functional interaction between tumor, stroma and lymphoid cells. Relative intratumoral transcriptomic homogeneity in one sample was associated with long-term overall survival, however other domains of heterogeneity were not evaluable with this technique.”

7) The limitations posed by spatially analyzing only a single small lesion from a single patient should be addressed.

This is indeed a relevant limitation, and we have added the following text to the Discussion:

“Our findings of extensive immunogenomic heterogeneity at the intra-tumoral level are inherently limited by detailed multi-platform profiling of a single lesion,

thus it is difficult to determine how typical the observed extent of heterogeneity is to broader patient populations, particularly those having differing clinical scenarios and treatment outcomes. Nonetheless, considering that sub-clonal variation has now been described in numerous tumor types, these findings serve to highlight the potential sensitivity of the immune microenvironment to local factors, including tumor genomic features that appear to have functional impact on local tumor immunity. Through analyses of several clinical datasets we found certain immunogenomic features from our deeply profiled tumor to have meaningful correlates in additional cohorts of patient samples, but additional studies are clearly required to refine these inferences towards therapeutically manipulable strategies.”

Minor:

a. lines 238-241: it would be helpful if low-minimal and moderate-high immune cell content thresholds were defined.

We agree with the reviewer that additional detail on this point would be of value, and a related question was raised by reviewer 1. In the context of immune infiltration of tumor masses, even digitally-quantified IHC requires a degree of manual curation to account for distribution of marker staining (eg: CD45) and for this specific study, consensus estimates using adjacent FFPE sections to interpolate immune content within the frozen sections used for genomic analyses. Thus immune cell content determination was at best semi-quantitative which is why we chose to represent this primarily as a categorical variable. We have updated the methods text to describe this better; essentially, we evaluated CD45-LCA positive cell density using digital image quantification using the Aperio ImageScope software. CD45 density was then categorized as low, medium or high by binning into lower, middle, or upper tertiles considering all regions analyzed. However, before arriving at a final semi-quantitative categorization as having focal, low, moderate or high immune infiltration, several factors required consideration including between-slide variation in staining efficiency, spatial distribution of immune cell content (focal, broad, intra/peri-/extra-tumoral), and a consensus estimate for genomic frozen slices by considering the adjacent FFPE sections on both sides for which essentially the higher immune content was assigned precedence.

b. Methods: the versions of all bioinformatics tools should be specified (eg MuTect, NetMHCpan, etc) for reproducibility and because outputs often vary slightly between versions.

We have updated the methods section to include these details.

c. line 488: best practices for removing duplicates, indel removal and recalibration should be referenced.

A citation has been added.

d. In addition to the 1x predicted neoantigen SNV shared across all tumor samples, it would be interesting to know how the remainder of the neoantigen repertoire is distributed across tumor subregions and how they are associated with the distribution of T-cell clonotypes; with the noted limitation that MHC binding affinity algorithms are severely handicapped.

Neoantigen data are derived from whole exome sequencing and thus are only available for 4 of the sub-regions profiled from lesion 2. From the data we have presented it is clear that there were no other universally-observed (present) neoantigens at least within these evaluable regions. Given the restricted number of sub-regions for which comprehensive neoantigen data are available, it is unfortunately not feasible to model the association of neoantigen repertoire with T cell repertoire in any meaningful way. As the reviewer points out, MHC binding affinity algorithms are subject to numerous well-known limitations, and additionally such modeling

would also be unable to consider (reliably) the local availability of 'best' MHC type to present any given predicted neoantigen, whether the necessary antigen presentation machinery is indeed expressed in order for a cognate interaction with T cells to occur, and whether any given T cell clonotype (based solely on V β chain sequence without known T cell phenotype or TCR alpha chain information) could indeed recognize any given neoantigen.

Reviewer #3 (Remarks to the Author): Expertise in genomics

The manuscript by Mitra et al. reports a very deep characterization of a single patient by performing immunogenomic analyses on 67 intra-tumor sub-regions of a PD-1 inhibitor resistant tumor, and 2 additional melanoma metastases. The work is very well presented and would be of great interest to the readership. The paper has some tantalizing findings, especially those related to chr 7 gain which were explored in a larger dataset.

Comments/questions to the authors:

1. Many of the conclusions from the paper are from a single individual, who was a long-term survivor who was heavily treated. The n of 1 is a potential weakness. Inclusion in the discussion of whether this sample is really representative of other cases needs to be included.

We agree that this is a potential limitation which was also raised by reviewer #2 (question 7), thus we have added the following text to the Discussion:

“Our findings of extensive immunogenomic heterogeneity at the intra-tumoral level are inherently limited by detailed multi-platform profiling of a single lesion, thus it is difficult to determine how typical the observed extent of heterogeneity is to broader patient populations, particularly those having differing clinical scenarios and treatment outcomes. Nonetheless, considering that sub-clonal variation has now been described in numerous tumor types, these findings serve to highlight the potential sensitivity of the immune microenvironment to local factors, including tumor genomic features that appear to have functional impact on local tumor immunity. Through analyses of several clinical datasets we found certain immunogenomic features from our deeply profiled tumor to have meaningful correlates in additional cohorts of patient samples, but additional studies are clearly required to refine these inferences towards therapeutically manipulable strategies.”

2. Was the degree of genomic heterogeneity (SNV and CNA) across the heavily pre-treated tumor surprising? Would you expect more clonal selection in such a pre-treated sample?

This is an excellent point and was also relevant to a question from reviewer 1. We definitely agree that the degree of genomic ITH observed in this particular tumor could be influenced by prior treatment exposures, which in this case includes targeted agents and prior immunotherapy. One might expect more clonal selection as a result of prior therapies, however this is predicated on at least partial efficacy of prior therapies in order for there to have been treatment susceptible subclones capable of being selected against. In this case, prior treatment exposures lacked significant clinical efficacy. Additionally, periods of tumor cell dormancy, especially in micrometastatic deposits prior to becoming clinically evident lesions, might confer relative resistance to systemic therapies, but essentially cannot be studied in human subjects as they are by definition not evident except in retrospect.

Considerations of efficacy aside, due to inherent differences in mechanism of action, distinct therapeutic agents likely also have distinct conditions under which their selective pressures are active. For example, certain modalities of therapy might be expected to have a stronger selective influence on cells that are actively proliferating at the time (e.g.: MAPK blockade)

whilst the selective pressure exerted by immunotherapies might be more dependent on expression of immunogenic antigens at the time which may have little to do with other cell state features like proliferation. Hence, when multiple therapies are sequenced, the net effect on endpoint genomic heterogeneity may be very difficult to predict and would not necessarily be expected to lead to an inevitable step-wise increase in tumor cell clonality.

3. Does the RNAseq indicate which genes are being affected by the copy number changes at chromosome 10 (tumor margin events)?

Of the differentially-expressed genes comparing chr10 CNA loss vs. stable samples 17/39 (44%) were genes located on chromosome 10 itself, but interestingly all were relatively highly-expressed with low fold-changes (all less than 2-fold) in contrast to differentially-expressed genes on other chromosomes which were generally of lower abundance but more differentially expressed between the chr10 copy number states. We have added the following text to the CNA section and included results as a supplementary table:

“Nearly half (17/39, 44%) of the differentially-expressed genes associated with chromosome 10 copy number losses were relatively highly-expressed by low fold-change genes located on chromosome 10 itself, but more pronounced changes (at fold-change level) were observed in differentially-expressed genes located on other chromosomes, such as MT1B (chr16), TNNT3 (chr11) and MUC12 (chr7) and RPS6KA6 (chrX).”

4. The ch7 finding is intriguing. Do the low counts of immune cells (from RNAseq) within chr7 regions correlate with IHC assessment of these regions? Even though immune cell counts were low, were the proportion of immune cells similar to other sites? What genes were in the ‘positive neutrophil activation’ and ‘negative neutrophil activation’ signatures?

In general, transcriptomic deconvolution of immune populations demonstrated broad representation of immune cells, when present. Therefore, evaluating immune cell content by CD45LCA positivity on IHC, chromosome 7 gain sub-regions had variable densities, ranging from 21% to 242% of the overall average CD45+ density. The apparent discrepancy between transcriptomic and IHC results is not unexpected and indeed highlights several pertinent points: 1) as described previously, transcriptomic data and IHC data were not generated from exactly the same pieces of tissue; rather estimates of immune cell densities had to be inferred by using adjacent IHC slices and may thus not be entirely accurate, 2) different methodologies have underlying technical considerations that warrant caution when considering cross-methodology comparisons, 3) absolute IHC densities are not ‘normalized’ to overall cell density (including tumor cell density), thus the density data obtained from IHC does not necessarily directly compare to the proportional densities produced by many transcriptomic deconvolution tools, and debate remains regarding the reliability of the output when some of these tools are run in “absolute” mode, 4) emerging evidence implicates individual cell-cell spatial relationships between immune and tumor cells (e.g.: proximity, Gide et al, Oncoimmunology DOI: 10.1080/2162402X.2019.1659093) as highly relevant to treatment outcomes, which is inherently not captured by simple density data.

With respect to immune cell proportions (relative) within these sites, there was extensive variation throughout the sub-regions, as evidenced by MCP-Counter and CIBERSORT immune deconvolution methods, including these sites (relative to other sites).

The genes in the relevant neutrophil activation (positive/negative) signatures from GO are provided in Supplementary Table 8.

The inclusion of the TCGA supports a role for chr7. Can the TCGA RNAseq also undergo deconvolution to see if there was a similar count or proportion of immune cells in the TCGA data?

As requested by reviewer 2, we have now performed CIBERSORT immune population deconvolution on the TCGA samples, which revealed relatively higher neutrophil signatures in chr7 stable tumors. These data are now presented in Supplementary Figure 8.

5. The authors show that the neutrophil signature is associated with response to immunotherapy in public datasets. Was chr7 gain also associated with response to checkpoint blockade?

We agree that this is a very pertinent question, however the publicly available immunotherapy-treated datasets do not have readily available (if at all) genomic data from which copy number variation can be appropriately assessed. Additionally, as this was not a focus of these studies, and CNV data generation is non-trivial, we are unable to perform this analysis in the current study but agree that it would be of compelling interest to evaluate this in future work.

6. There is a large amount of omic work (RNAseq, DNaseq, methylation, TCR) from many small samples that may have been technically challenging. Can the authors summarise the data quality, ie:

a. The nucleotide yield must have been sufficient for these analyses, what was the yield of DNA and RNA for each site and the quality of the nucleotides (eg RIN score)?

RNAseq was performed on samples with a minimum RNA integrity number (RIN) of 5.5 except for two cases (6A10 and 8A3) with RINs greater than 3. Additionally, a minimum of 700ng of RNA were required for all samples undergoing RNAseq. These details have been added to the methods section "Nucleic acid extraction".

b. What read depth was the RNA sequenced?

Read depth, in terms of total library size (raw counts) per sample averaged 15 million reads, and has been indicated in Supplementary Table 1.

c. The TCR sequencing results seem highly variable across the tumor. Could this be due to under sampling the TCR at each site?

We considered TCR sequencing depth in terms of the number of unique templates identified per sample (provided in Supplementary Table 1) to ensure that TCR sequencing itself was adequate, and thus that any variation in TCR repertoire between regions was not simply an artifact of regional under-sampling (i.e.: QC failure). We expected that the impetus and ability of T cells to infiltrate the tumor would vary from region to region according to numerous factors such as local antigenic stimuli, cytokine gradients, and vascular accessibility, thus we were not surprised to identify significant variation in TCR sequencing results across the tumor.

7. To support the conclusion that the common epitope is driving a persistent T-cell response, is it possible to validate the immunogenicity of the ZDNNC17 p.H507Y neoantigen in vitro?

This is indeed possible, and we sought to test the immunogenicity of this neoantigen using established peptide stimulation assays as previously described (Pollack et al, 2014, J Immunother Cancer, <https://www.ncbi.nlm.nih.gov/pubmed/25317334>). We have now completed these experiments and the results have been incorporated into the manuscript, and detailed methods added.

Briefly, to evaluate the potential in vitro immunogenicity of the ZDNNC17 p.H507Y neoantigen, we synthesized 12 overlapping candidate 9-mer peptides spanning the neoantigenic point mutation and used these peptides to elicit CD8+ T cell responses from HLA-A*0301 donor PBMC. HLA-A*0301-transfected K562 (A3-K562) cells were pulsed with each candidate peptide

and used as antigen presenting cells for stimulating CD8 T cells from two HLA-A*0301 in a 48 well plate format. Three rounds of PBMC stimulation with peptide pulsed A3-K562 were performed at 7-day intervals with IL-21 present in the culture medium throughout. After three rounds of stimulation, an aliquot of 100,000 cells from each well was co-cultured overnight with peptide pulsed K562 cells at a ratio of 10:1 to assay for antigen specific T cells using a flow cytometry based intracellular IFN- γ production assay. The percentage of background IFN- γ positive cells was determined by the response of PBMC co-cultured with non-pulsed K562 cells and the responses were registered as positive if the proportion of T cells producing IFN- γ in response to stimulation with ZDNNC17 p.H507Y derived peptide was ≥ 2 -fold greater than the background proportion of IFN- γ + CD8 T cells.

Compared with donor PBMC co-cultured with non-pulsed A3-K562 cells, we observed elevated IFN- γ production by HLA-A*0301 donor CD8+ T cells for several peptides, in particular 4, 6, 7, 9, 11 and 12 (Table) . Expressed as fold-increase above baseline, peptides 4 and 12 appear to induce the most robust responses, at an average of 3-5-fold increase above background levels (Figure). Peptide 4 and Peptide 12 thus represent immunogenic epitope candidates on the basis of this analysis.

Table:

	Donor 1	Donor 2
Peptide 4	1.9	0.74
Peptide 6	1.33	0.46
Peptide 7	0.93	0.95
Peptide 9	1.45	0.63
Peptide 11	1.13	0.70
Peptide 12	1.66	0.64
unpulsed K562	0.39	0.24

Figure:

Figure legend: Potential epitope specific CD8+ T cells in PBMC were stimulated three times at 7-day intervals by co-culturing with HLA-A*0301 expressing K562 (A3-K562) pulsed with overlapping peptides (Peptides 1-12) in the presence of IL-21. Epitope specificity of sensitized CD8+ T cells was determined by co-culturing cells with peptide pulsed K562 overnight and performing a standard intracellular IFN- γ production assay. Cells co-cultured with non-pulsed K562 cells served as Background (negative) controls. Treatment with PMA and ionomycin served as a positive control for the assay. The table (at left) lists the highest responding peptide candidates as % IFN- γ + CD8+ T cells among total CD8+ T cells (baseline of 0.39 and 0.24% from Donors 1 and 2, respectively). Fold-increase in peptide reactivity above baseline (1.0) is shown in the Figure (at right).

Minor comments

- It would be helpful to include more information on the figures or in the legends in terms of axis labels and legends so the values are clear, for example Fig 5A and B what is the value and how were the scores generated, also what is the value of 'size' in Fig 5C etc. Fig 1E include a scale indicating level of CNA.

We have added additional clarifications to figure legends as requested.

- Ensure all methods described were used in the paper. For example the single cell methods include assessment of CNA using CONICS, but I did not see inclusion of this data into the paper.

We apologize for this oversight – we did not perform CONICS analysis in this study and have accordingly removed it from the methods.

REVIEWERS' COMMENTS:

Reviewer #1 (Remarks to the Author):

Thanks for the detailed responses. The authors have addressed all of my major concerns and clarified them in the manuscript.

Reviewer #3 (Remarks to the Author):

The response is very detailed, the new analyses are good and all of my questions/comments have been addressed.

Reviewer #4 (Remarks to the Author): Replacement for Reviewer #2

As requested by the editor and appropriate I will restrict my comments primarily to the concerns of reviewer 2 as this manuscript has already been through the peer review and revision process. Overall, the manuscript I believe represents an extremely impressive effort to comprehensively and systematically analyze intratumoral heterogeneity both from an immune perspective and tumor genomics. While the data is limited to a single patient, and primarily a single tumor, the data and analysis is robust and will be of significant interest to the scientific community. The authors have performed additional analyses to address the reviewers comments which I believe have improved the manuscript and likely only requires minor additional changes.

1. The care that was taken to precisely delineate how the samples were processed and selected for sequencing is critical information as pointed out by reviewer 2 and the authors have made efforts to expand this further. The reviewer's concerns in my opinion have been satisfactorily addressed.

2. The authors demonstrate an association between CN gains in chromosome 7 and altered immune composition and argue that neutrophil activation may be an associated mechanism. The authors then correlate these findings in TCGA data as well as immunotherapy treated cohorts. As discussed by the reviewer and the authors in the rebuttal neutrophil accumulation can and is likely associated with necrosis and thus may ultimately drive any bulk level analysis performed in TCGA or IO treated cohorts (e.g. chr 7 associated with intrinsic bad biology tumor, increased proliferation/necrosis and worse prognostic association in survival or response analyses).

While the authors note the differences in neutrophil infiltration and activation in the rebuttal and these associations are clear in their paired analysis of individual regions where samples were available, these associations on bulk analyses are still unclear. The additional analyses using CIBERSORT data from these samples is helpful and encouraging although the precision of CIBERSORT data for neutrophils may not be extremely high. Any additional analyses such as CNA association in the IO cohorts falls outside the scope of this study and would require significant effort/would likely be limited by data quality as suggested by the authors.

The authors have gone to considerable effort and performed additional analyses to support their assertions but in my opinion language used in the abstract and manuscript text regarding neutrophil association should be tempered and a comment should likely be included in the discussion regarding possible association with necrosis, etc.

3-7. The authors response is satisfactory.

Minor comment

For the Riaz et al dataset it is important to note which samples were analyzed for the RNAseq analysis as paired samples before and on treatment were performed. Based on the n=56, this likely represents on-treatment samples but this should be confirmed and included in the manuscript.

REVIEWERS' COMMENTS:

Reviewer #1 (Remarks to the Author):

Thanks for the detailed responses. The authors have addressed all of my major concerns and clarified them in the manuscript.

Reviewer #3 (Remarks to the Author):

The response is very detailed, the new analyses are good and all of my questions/comments have been addressed.

Reviewer #4 (Remarks to the Author): Replacement for Reviewer #2

As requested by the editor and appropriate I will restrict my comments primarily to the concerns of reviewer 2 as this manuscript has already been through the peer review and revision process.

Overall, the manuscript I believe represents an extremely impressive effort to comprehensively and systematically analyze intratumoral heterogeneity both from an immune perspective and tumor genomics. While the data is limited to a single patient, and primarily a single tumor, the data and analysis is robust and will be of significant interest to the scientific community. The authors have performed additional analyses to address the reviewers comments which I believe have improved the manuscript and likely only requires minor additional changes.

1. The care that was taken to precisely delineate how the samples were processed and selected for sequencing is critical information as pointed out by reviewer 2 and the authors have made efforts to expand this further. The reviewer's concerns in my opinion have been satisfactorily addressed.

2. The authors demonstrate an association between CN gains in chromosome 7 and altered immune composition and argue that neutrophil activation may be an associated mechanism. The authors then correlate these findings in TCGA data as well as immunotherapy treated cohorts.

As discussed by the reviewer and the authors in the rebuttal neutrophil accumulation can and is likely associated with necrosis and thus may ultimately drive any bulk level analysis performed in TCGA or IO treated cohorts (e.g. chr 7 associated with intrinsic bad biology tumor, increased proliferation/necrosis and worse prognostic association in survival or response analyses).

While the authors note the differences in neutrophil infiltration and activation in the rebuttal and these associations are clear in their paired analysis of individual regions where samples were available, these associations on bulk analyses are still unclear. The additional analyses using CIBERSORT data from these samples is helpful and encouraging although the precision of CIBERSORT data for neutrophils may not be extremely high. Any additional analyses such as CNA association in the IO cohorts falls outside the scope of this study and would require significant effort/would likely be limited by data quality as suggested by the authors.

The authors have gone to considerable effort and performed additional analyses to support their assertions but in my opinion **language used in the abstract and manuscript text regarding neutrophil association should be tempered and a comment should likely be included in the discussion regarding possible association with necrosis, etc.**

Additional wording has been included in the abstract, results and discussion to note the potential confounding association with necrosis/reactive inflammatory infiltrate, as requested.

3-7. The authors response is satisfactory.

Minor comment

For the Riaz et al dataset it is important to note which samples were analyzed for the RNAseq analysis as paired samples before and on treatment were performed. Based on the n=56, this likely represents on-treatment samples but this should be confirmed and included in the manuscript.

This has been clarified.